# LEARNING OBJECT-CENTRIC LATENT DYNAMICS FOR REINFORCEMENT LEARNING FROM PIXELS

## ABSTRACT

Learning a latent dynamics model provides a task-agnostic representation of an agent's understanding of its environment. Leveraging this knowledge for model-based reinforcement learning holds the potential to improve sample efficiency over model-free methods by learning inside imagined rollouts. Furthermore, because the latent space serves as input to behavior models, the informative representations learned by the world model facilitate efficient learning of desired skills. Most existing methods rely on holistic representations of the environment's state. In contrast, humans reason about objects and their interactions, forecasting how actions will affect specific parts of their surroundings. Inspired by this, we propose *Slot-Attention for Object-centric Latent Dynamics (SOLD)*, a novel algorithm that learns object-centric dynamics models in an unsupervised manner from pixel inputs. We demonstrate that the structured latent space not only improves model interpretability but also provides a valuable input space for behavior models to reason over. Our results show that *SOLD* outperforms DreamerV3, a state-of-the-art model-based RL algorithm, across a range of benchmark robotic environments that evaluate for both relational reasoning and low-level manipulation capabilities.

## 1 INTRODUCTION

Advances in reinforcement learning (RL) have showcased the ability to learn sophisticated control strategies through interaction, achieving superhuman performance in domains ranging from board games (Silver et al., 2016) to drone racing (Kaufmann et al., 2023). While these approaches excel in settings where explicit models of the environment are available or abundant data can be collected, learning complex control tasks in a sample-efficient manner remains a significant challenge. Model-based RL (MBRL) has emerged as a promising approach to address this limitation by constructing models of the environment's dynamics. Notably, the Dreamer framework (Hafner et al., 2019; 2020; 2023) has demonstrated improved sample efficiency over model-free baselines by learning behaviors solely through imagined rollouts.

While these research efforts have produced world models capable of accurately predicting the dynamics of visual tasks, they rely on a holistic representation of the environment's state. In contrast, humans perceive the world by parsing scenes into individual objects (Spelke, 1990), anticipating how their actions will influence specific components of their surroundings. This ability is crucial in complex tasks that require reasoning about multiple objects and their interactions, which is common in robotic manipulation. Learning structured representations of an environment through objects introduces a powerful inductive bias. The learned representations not only enhance the interpretability of the dynamics prediction but also foster decision-making by providing a structured input space for behavior models to reason over. Despite these advantages, the integration of object-centric representations and world models remains largely underexplored. To the best of our knowledge, no prior work has introduced a method that performs object-centric model-based RL directly from pixels.

To address the limitations of holistic representations in model-based RL, we propose *Slot-Attention for Object-centric Latent Dynamics (SOLD)*, a novel algorithm that leverages structured, object-centric states within the latent space of its world model. Our method introduces two key innovations to the model-based RL framework. The first contribution is a dynamics model that predicts future frames in terms of their slot representation. To achieve this, we extend OCVP (Villar-Corrales et al., 2023) into an action-conditional model by projecting action commands into the slot

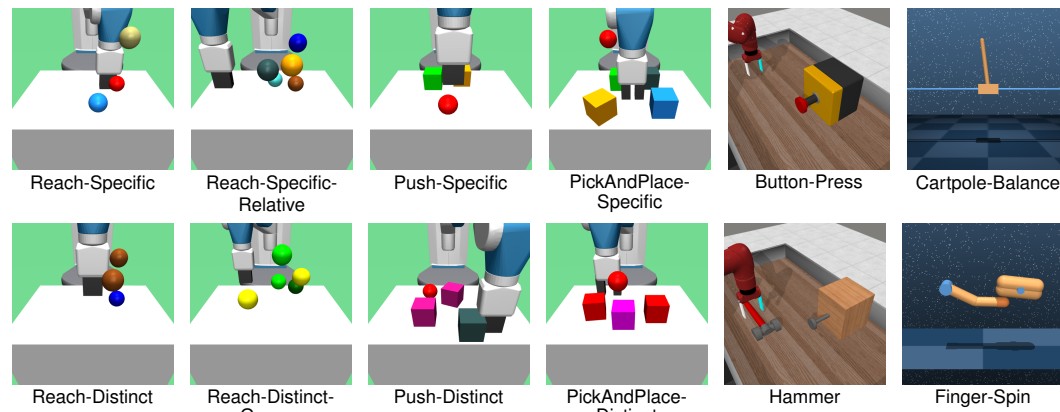

Figure 1: We evaluate SOLD on diverse visual environments from (left to right) our multi-object robotic control benchmark, Meta-World (Yu et al., 2019), and DM-Control (Tassa et al., 2018).

dimension. A transformer-based (Vaswani et al., 2017) transition model processes the set of object-slots and actions to predict the subsequent frame. Notably, the dynamics model is trained solely from pixels through a loss on the reconstructions and slot representations of the predicted frames. The second key component is a transformer-based model backbone, which we call *Slot Aggregation Transformer*, that is used to make time-step-wise predictions from the history of object slots. Specifically, it serves as the architectural backbone to the reward, value, and action model, allowing us to perform model-based RL training on the basis of object-centric representations.

For systematic evaluation, we introduce a suite of visual robotics tasks, shown in Figure 1, that require varying levels of relational reasoning and manipulation capabilities. We perform an extensive comparison on this benchmark, demonstrating that our method achieves superior performance to the state-of-the-art MBRL algorithm DreamerV3 (Hafner et al., 2023). Furthermore, we apply SOLD to tasks from two RL benchmarks that were not designed to be object-centric, providing evidence of the generalizability of our framework. In summary, we make the following contributions:

- We introduce SOLD, which is to the best of our knowledge, the first object-centric model-based RL algorithm that learns entirely from pixel inputs.
- Our method outperforms DreamerV3 across a range of visual robotics environments, excelling in tasks that require both relational reasoning and low-level manipulation skills.
- We overcome limitations of prior object-centric methods in RL. Namely, we show that our object-centric encoder-decoder module can be adapted to state distributions vastly different from those seen under a random policy during pre-training – an essential capability for solving many complex tasks. Additionally, we show its generalization potential to environments not explicitly designed for object-centric reasoning.

## 2 BACKGROUND

**Slot Attention for Video (SAVi)**   SOLD employs SAVi (Kipf et al., 2022), an encoder-decoder architecture with a structured bottleneck composed of $N$ permutation-equivariant object embeddings denoted as slots, in order to recursively parse a sequence of video frames $\boldsymbol{o}_{0:\tau} = \{\boldsymbol{o}_0, ..., \boldsymbol{o}_\tau\}$ into their object representations $\boldsymbol{Z}_{0:\tau} = \{\boldsymbol{Z}_0, ..., \boldsymbol{Z}_\tau\}, \boldsymbol{Z}_t \in \mathbb{R}^{N \times D_z}$. At time $t$, SAVi encodes the input video frame $\boldsymbol{o}_t$ into a set of feature maps $\boldsymbol{F}_t \in \mathbb{R}^{L \times D_h}$, where $L$ is the size of the flattened grid (i.e. $L = \text{width} \cdot \text{height}$), and uses Slot Attention (Locatello et al., 2020) to iteratively refine the previous slot representations conditioned on the current features. Slot Attention performs cross-attention between the slots and image features with the attention coefficients normalized over the slot dimension, thus encouraging the slots to compete to represent feature locations:

$$\boldsymbol{A} \doteq \operatorname*{softmax}_{N}\left(\frac{q(\boldsymbol{Z}_{t-1}) \cdot k(\boldsymbol{F}_t)^T}{\sqrt{D}}\right) \in \mathbb{R}^{N \times L}, \qquad (1)$$

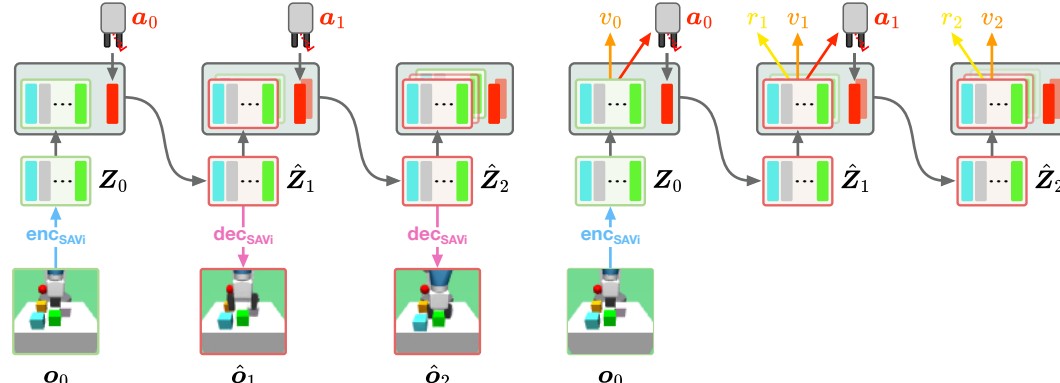

(a) World Model Learning: SAVi encodes images $o_t$ into slots $Z_t$, which are predicted by the dynamics model given history of slots and actions. We reconstruct the images and compute their actual slot representation to shape the dynamics prediction.

(b) Behavior Learning: The actor and critic are trained via imagined rollouts in the latent space of the world model. Trajectories start after $S$ seed frames (visualized for $S = 1$) and predict forward with actions sampled from the actor network.

Figure 2: *SOLD* is trained by concurrently making the world model consistent with replayed experiences and learning behaviors through latent imagination.

where $q$ and $k$ are learned linear mappings to a common dimension $D$. The slots are then independently updated via a shared Gated Recurrent Unit (Cho, 2014) (GRU) followed by a residual MLP:

$$Z_t \doteq \text{MLP}(\text{GRU}(\boldsymbol{A} \cdot v(\boldsymbol{F}_t), Z_{t-1})) \ \ \text{with} \ \ \boldsymbol{A}_{n,l} \doteq \frac{\boldsymbol{A}_{n,l}}{\sum_{i=0}^{L-1} \boldsymbol{A}_{n,i}}, \tag{2}$$

where $v$ is a learned linear projection. The steps described in Equations 1 and 2 can be repeated multiple times with shared weights to iteratively refine the slots and obtain an accurate object-centric representation of the scene. Finally, SAVi independently decodes each slot of $Z_t$ into per-object images and alpha masks, which can be normalized and combined via weighted sum to render video frames. SAVi is trained end-to-end in a self-supervised manner with an image reconstruction loss.

**OCVP** Our dynamics model builds on OCVP (Villar-Corrales et al., 2023) in order to autoregressively predict future object slots conditioned on past object states. OCVP is a transformer-encoder model that decouples the processing of object dynamics and interactions, thus leading to interpretable and temporally consistent object predictions while retaining the inherent permutation-equivariant property of the object slots. This is achieved through the use of two specialized self-attention variants: *temporal attention* updates a slot representation by aggregating information from the corresponding slot up to the current time step, without modeling interactions between distinct objects, whereas *relational attention* models object interactions by jointly processing all slots from the same time step.

## 3 SLOT-ATTENTION FOR OBJECT-CENTRIC LATENT DYNAMICS

We propose *Slot-Attention for Object-centric Latent Dynamics (SOLD)*, a method that combines model-based RL akin to the Dreamer framework (Hafner et al., 2019; 2020; 2023) with object-centric representations. The three core components of our algorithm are: the *object-centric world model*, which predicts the effects of actions on the environment, the *critic*, which estimates the value of a given state, and the *actor* that selects actions to maximize this value.

Figure 2 gives an overview of the training process. The world model operates on structured latent states by splitting the environment into its constituent objects and then composing future frames via the predicted states of these individual components. Specifically, we pretrain a SAVi encoder-decoder model (Kipf et al., 2022) on random sequences from the environment to extract object-centric representations. After pretraining, all components of the world model are trained jointly using replayed experiences from the agent's interaction with the environment. The actor and critic

are trained on imagined sequences of structured latent states. We execute actions sampled from the actor model in the environment and append the resulting experiences to the replay buffer. Detailed explanations of world model learning and behavior learning are provided in Sections 3.1 and 3.2, respectively.

## 3.1 WORLD MODEL LEARNING

World models compress an agent's experience into a predictive model that forecasts the outcomes of potential actions. By simulating rollouts within the internal model, agents can learn desired behaviors in a sample-efficient manner. When the inputs are high-dimensional images, it is helpful to learn compact state representations, enabling prediction within this latent space. This type of model, called latent dynamics model, allows for efficient prediction of many latent sequences in parallel.

Most prior works rely on generating a single, holistic representation of the environment's state, which contrasts with findings from cognitive psychology. Humans perceive scenes as compositions of objects (Spelke, 1990) and reason about how their actions affect distinct parts of their environment. Furthermore, environment dynamics can be compactly explained in terms of objects and their interactions Battaglia et al. (2016). Therefore, we propose to structure the latent space by decomposing visual environments into their constituent parts.

**Components** To create a world model that operates on object-centric latent representations, we build on top of OCVP (Villar-Corrales et al., 2023). We begin by pretraining SAVi on a dataset of $10^6$ frames from random episodes. Having a sufficiently large initial dataset is crucial for meaningful object-centric representations to emerge. We do not freeze the pretrained encoder-decoder models, allowing slots to adapt to novel configurations that do not occur during random pre-training. The sequence of object slots $\boldsymbol{Z}_{0:t}$ alongside the action commands $\boldsymbol{a}_{0:t}$ serve as inputs to our transformer-based dynamics model which predicts the slot representation of the next frame $\hat{\boldsymbol{Z}}_{t+1}$:

$$
\begin{aligned}
\text{Encoder:} \quad & \boldsymbol{Z}_t = e_\eta(\boldsymbol{o}_t), \\
\text{Decoder:} \quad & \hat{\boldsymbol{o}}_t = d_\eta(\boldsymbol{Z}_t), \\
\text{Dynamics model:} \quad & \hat{\boldsymbol{Z}}_{t+1} = p_\psi(\boldsymbol{Z}_{0:t}, \boldsymbol{a}_{0:t}), \text{ and} \\
\text{Reward predictor:} \quad & \hat{r}_t \sim p_\zeta(\hat{r}_t \mid \boldsymbol{Z}_{0:t}).
\end{aligned}
\tag{3}
$$

**Object-centric dynamics learning** For the dynamics model, we follow the sequential attention pattern proposed in Villar-Corrales et al. (2023), which disentangles relational and temporal attention to decouple the processing object dynamics and interactions. During training, we provide the slot representation of $S$ seed frames as context. We append the predictions to the context and apply this process in an autoregressive manner to predict the subsequent $T$ frames. We do not employ teacher forcing so that the dynamics model learns to handle its own imperfect predictions. To shape the predicted representations, we reconstruct the subsequent frame $\hat{\boldsymbol{o}}_{t+1}$ and extract the SAVi representations of the actual frame $\boldsymbol{Z}_{t+1}$ to compute the hybrid dynamics loss:

$$
\mathcal{L}_{\text{dyn}}(\psi) \doteq \sum_{t=S}^{S+T-1} \left[ \underbrace{\left\| \hat{\boldsymbol{Z}}_t - e_\eta(\boldsymbol{o}_t) \right\|_2^2}_{\text{Joint embedding}} + \underbrace{\left\| \hat{\boldsymbol{o}}_t - \boldsymbol{o}_t \right\|_2^2}_{\text{Reconstruction}} \right].
\tag{4}
$$

For all loss terms, we specify the parameter group that is being optimized and omit stop-gradients for other models to avoid cluttering the notation.

**Reward model learning** The reward predictor solves a regression problem where the prediction depends on the slot representations but is not directly tied to any single slot. To address this, we introduce the *Slot Aggregation Transformer (SAT)* as an architectural backbone, which introduces output tokens and a variable number of register tokens for all time-steps. Register tokens, recently shown to enhance computation in vision transformers (Darcet et al., 2024), can aid computation when processing a set of inputs to produce a singular output. To encode the position information, we adopt ALiBi (Press et al., 2022) in place of absolute position encoding. ALiBi introduces linear biases directly into the attention scores, effectively encoding token recency. This approach helps to

generalize to sequences longer than those seen during training. A detailed description of the SAT can be found in Section C.3. To efficiently represent a wide range of reward values, we avoid directly predicting a scalar reward. Instead, the MLP head $f_\zeta$ outputs logits of a softmax distribution over $K$ exponentially spaced bins $b_i$. The predicted reward can then be computed as the expectation over these bins:

$$\boldsymbol{b} \doteq \mathrm{symexp}([-20, ..., +20]), \qquad \hat{r}_t \doteq \mathrm{softmax}(f_\zeta^{\mathrm{MLP}}(\boldsymbol{h}_t))^T \boldsymbol{b}, \qquad (5)$$

where $\boldsymbol{h}_t$ are the output tokens after being processed by the SAT backbone. To formulate the loss, the true reward $r_t$ is first transformed using the symlog function (Webber, 2012) and then encoded via a two-hot encoding strategy (Bellemare et al., 2017; Schrittwieser et al., 2020). The model is trained to maximize the log-likelihood of the two-hot encoded reward distribution under the predicted distribution:

$$\mathcal{L}_{\mathrm{rew}}(\zeta) \doteq - \sum_{t=0}^{T-1} \log p_\zeta(r_t \mid \boldsymbol{Z}_{0:t}). \qquad (6)$$

## 3.2 Behavior Learning

Our strategy of using the world model for behavior learning builds upon the Dreamer framework. At the core of this method lies the process of latent imagination, visualized in Figure 2b, which trains the actor and critic networks purely on imagined trajectories predicted by the world model. Since both the actor and critic operate on the latent state, they benefit from the structured representation learned by the world model. The architecture of both models mirrors that of the reward predictor, consisting of a SAT backbone that processes the slot histories followed by an MLP head:

$$\text{Actor:} \quad \boldsymbol{a}_t \sim \pi_\theta(\boldsymbol{a}_t \mid \boldsymbol{Z}_{0:t}), \qquad \text{Critic:} \qquad \hat{R}_t \doteq \mathrm{E}[v_\phi(\hat{R}_t \mid \boldsymbol{Z}_{0:t})]. \qquad (7)$$

**Critic learning** To account for rewards beyond the imagination horizon $T = 15$, the critic is trained to estimate the expected return under the current actor's behavior. Since no ground truth is available for these estimates, we compute bootstrapped $\lambda$-returns (Sutton & Barto, 2018), $R^\lambda$, via temporal difference learning. These returns integrate predicted rewards $\hat{r}$ and values $\hat{R}$ to form the target for the value model:

$$R_t^\lambda \doteq \hat{r}_{t+1} + \gamma \left( (1-\gamma)\hat{R}_{t+1} + \lambda R_{t+1}^\lambda \right) \quad \text{where} \quad R_T^\lambda \doteq \hat{R}_T, \qquad (8)$$

which is trained to minimize the resulting loss:

$$\mathcal{L}_{\mathrm{critic}}(\phi) \doteq - \sum_{t=0}^{T-1} \log v_\phi(R_t^\lambda \mid \boldsymbol{Z}_{0:t}). \qquad (9)$$

We decouple the gradient scale from value prediction through same approach as in the reward model, predicting a categorical distribution over exponentially spaced bins. To stabilize learning, we regularize the critic's predictions towards the outputs of an exponentially moving average (EMA) of its own parameters (Mnih et al., 2015; Hafner et al., 2023).

**Actor learning** The actor is optimized to select actions that maximize its expected return while encouraging exploration through an entropy regularizer. Its model architecture is similar to the critic and reward predictor, but instead of regressing a scalar value, it predicts the parameters of the action distribution. Specifically, the MLP head outputs the mean $\boldsymbol{\mu}_t$ and standard deviation $\boldsymbol{\sigma}_t$ to parameterize a normal distribution $\mathcal{N}(\boldsymbol{\mu}_t, \boldsymbol{\sigma}_t | \boldsymbol{Z}_{0:t})$ over possible actions. The trade-off in the actor's loss function weights expected returns with maintaining randomness in the actor outputs and is hence subject to reward scale and frequency of the current environment. To adapt to varying scales of value estimates across different environments, we use a normalization factor $\mathrm{scale}_V$:

$$\mathcal{L}_{\mathrm{actor}}(\theta) \doteq - \sum_{t=0}^{T-1} \frac{\hat{R}_t^\lambda}{\max(1, \mathrm{scale}_V)} + \eta \mathrm{H}(\mathcal{N}(\boldsymbol{\mu}_t, \boldsymbol{\sigma}_t \mid \boldsymbol{Z}_{0:t})), \qquad (10)$$

where the value normalization is computed via the EMA of the 5th and 95th percentile of the value estimates (Hafner et al., 2023):

$$\mathrm{scale}_V \doteq \mathrm{EMA}\left( \mathrm{Per}(\hat{R}_t^\lambda, 95) - \mathrm{Per}(\hat{R}_t^\lambda, 5), 0.99 \right). \qquad (11)$$

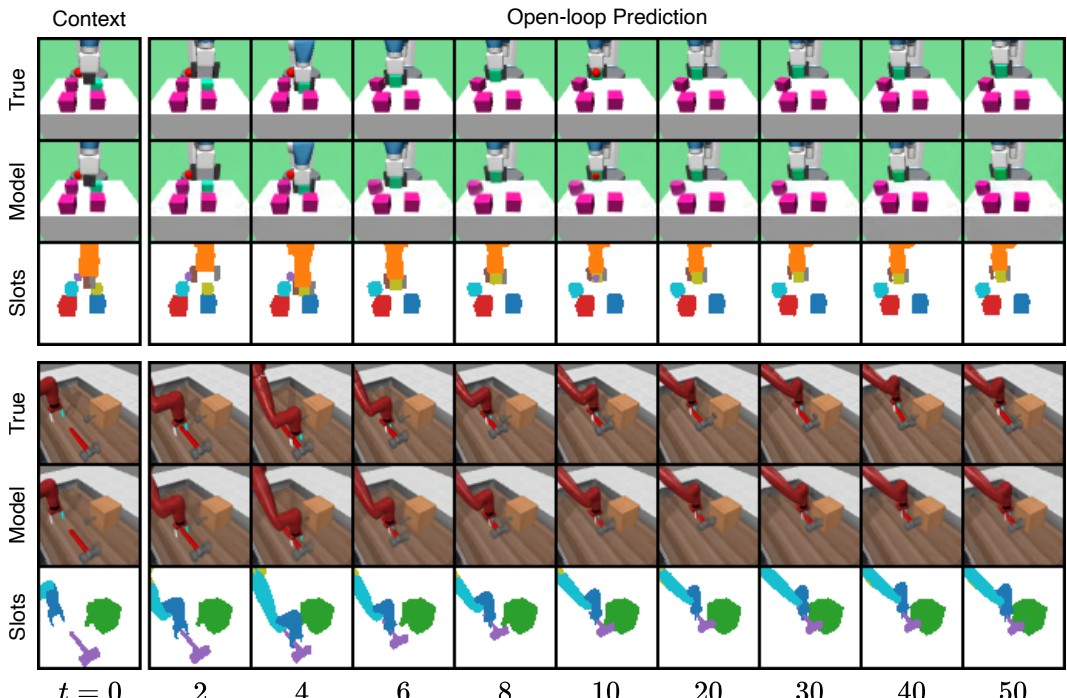

Figure 3: Open-loop predictions on the tasks *PickAndPlace-Distinct* (top) and *Hammer* (bottom). Starting from a single context frame, our model predicts the next 50 frames by propagating the individual slot representations forward without access to any intermediate images.

## 4 RESULTS

We evaluate SOLD on a suite of robotic manipulation tasks, designed to test for both complex relational reasoning as well as low-level manipulation capabilities. Further, we apply our method to environments that are not designed as object-centric tasks. Specifically, we test on two environments from Meta-World (Yu et al., 2019) and DM-Control (Tassa et al., 2018), respectively.

**Baselines** We design the experiments to achieve two main objectives: first, to specifically assess the impact of the object-centric paradigm in our method by comparing it to a baseline that replaces the object-centric encoder-decoder modules with a standard convolutional architecture (Ours w/o OCE); and second, to evaluate our approach against the best available competitor from the literature by benchmarking it against DreamerV3 (Hafner et al., 2023), a state-of-the-art MBRL algorithm known for its strong performance across a wide range of tasks. Here, we choose the 12 million parameter version, to match the parameter count of our own model. Further details about the baselines are provided in Appendix D.

**Environments** We introduce a suite of eight object-centric robotic control environments designed to test both relational reasoning and manipulation capabilities. These environments feature two types of problems: *Reach* tasks, where the agent must identify a target and move the end-effector to its location, and manipulation tasks (*Push* and *PickAndPlace*), where the agent identifies a target block and moves it to a designated goal. The action-space is 4-dimensional, where the first three components represent the desired movement direction of the end effector, and the fourth controls the gripper. On the *Reach* and *Push* tasks, commands to the gripper are ignored, with the gripper fixed in a closed configuration, as gripping is not required to solve these tasks. To test varying levels of relational reasoning difficulty, we design the following tasks:

- *Specific* The target object is red, with 0 to 4 distractor objects of random, distinct colors present in the scene.

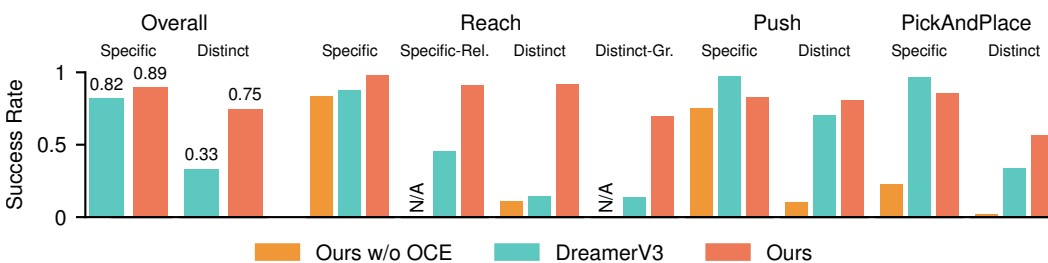

Figure 4: Final success rates across the eight evaluated environments. Overall, SOLD compares favorably to DreamerV3, showing slight improvements on the *Specific* tasks and a significant performance boost on the *Distinct* variants.

- *Distinct*  Inspired by the odd-one-out task in cognitive science (Crutch et al., 2009; Beatty & Vartanian, 2015), this task presents 3 to 5 objects, and the target is the one that differs in color from all the others.

For the *Reach* task, we also introduce two more challenging variants:

- *Specific-Relative*  Instead of reaching for the red object, the goal is to reach the reddest object, determined by the perceptual CIEDE2000 (Sharma et al., 2005) distance.

- *Distinct-Groups*  The environment always contains 5 targets, and the goal is to reach the one that appears only once.

On these two additional reach tasks, we reuse the SAVi models that were pre-trained for *Reach-Specific* and *Reach-Distinct*, respectively without modification. Further details about these environments are provided in Appendix E.

In addition to these tasks, we evaluate our approach on environments not originally designed for object-centric learning to investigate its generalizability. We include the *Button-Press* and *Hammer* tasks from the Meta-World benchmark (Yu et al., 2019), both featuring objects with complex shapes and textures, testing the model's ability to handle more diverse visual inputs. Finally, we assess our method on the *Cartpole-Balance* and *Finger-Spin* environments from the DM-Control suite (Tassa et al., 2018), which represent vastly different domains not typically associated with object-centric approaches. An overview of all studied environments is given in Figure 1.

### 4.1 OBJECT-CENTRIC DYNAMICS LEARNING

The object-centric representations learned by SAVi can be seen in the context-frames of Figure 3. The slots effectively decompose the visual scene, with most slots representing distinct objects and three slots capturing different parts of the respective robots. This part-whole segmentation demonstrates that the slots can meaningfully identify separate parts of a larger object, representing the gripper jaws in the first example and different parts of the kinematic chain of the Sawyer robot in the second. Notably, the sharp mask predictions show that each slot isolates information about the specific object it represents (see also Section F). This property is crucial for object-centric behavior learning, as it enables subsequent components to reason about task-relevant objects while ignoring irrelevant information.

Further, the open-loop predictions visualized in Figure 3 demonstrate that the object decomposition learned by the SAVi model is preserved throughout the prediction sequence. On the *PickAndPlace-Distinct* task, the movement of the robot and the blocks is predicted with high accuracy, showcasing the model's ability to capture complex physical interactions. Furthermore, the model effectively handles occlusions, as evidenced by the continued precise prediction of the spherical red target. In the second example, the slots capture the intricate shape of the hammer and nail-box. The predictions remain reliable over a long horizon, even during interactions between the robot and the hammer, and between the hammer and nail.

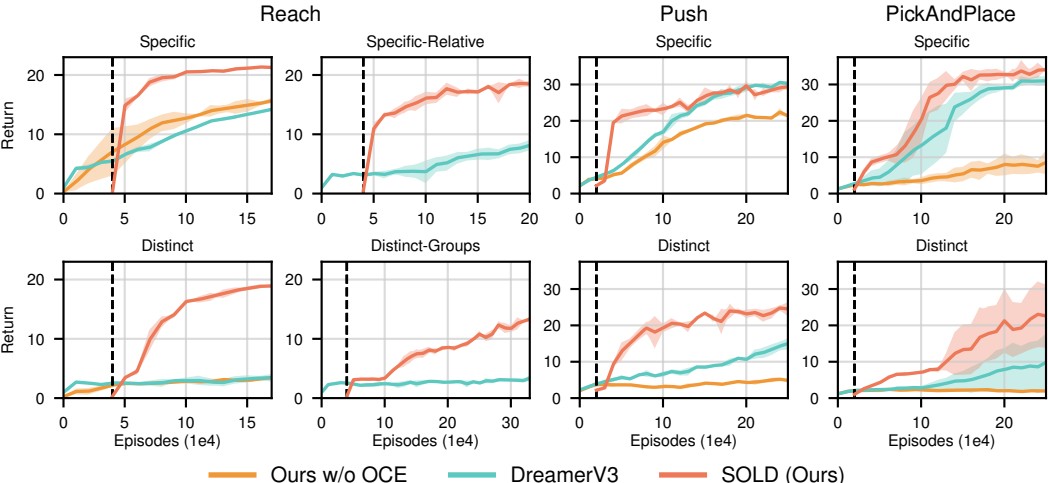

Figure 5: Achieved return over the training duration on our eight benchmark environments. The dotted vertical line indicates the offset of our method to account for the data used during pre-training.

## 4.2 BEHAVIOR LEARNING

To assess SOLD's performance across our suite of robotic control tasks, we train each method three times with different random seeds on each environment. The final success rate achieved by each method is shown in Figure 4. SOLD consistently outperforms the non-object-centric baseline, often by a significant margin, highlighting the effectiveness of object-centric representations for these problems. While the non-object-centric baseline is able to make progress on the *Specific* variants of the tasks, it struggles to perform the relational reasoning required to solve the *Distinct* versions. As a result, we decided excluded it from our evaluation of the two more advanced versions of the *Reach* tasks. When compared to the state-of-the-art method DreamerV3, our model shows competitive or superior performance. This advantage is particularly pronounced for tasks that require relational reasoning between objects, namely the *Distinct* variants and the *Specific-Relative* task.

These results confirm our hypothesis that learning a structured latent representation in the world model benefits downstream tasks that require object reasoning, which are common in robotics contexts. Beyond that, when examining the performance over the course of training, as shown in Figure 5, we observe additional advantages. Even after accounting for the samples used during pre-training, our method consistently outperforms the highly sample-efficient DreamerV3 baseline, learning quickly and achieving high returns with minimal experience. This enhanced sample efficiency is observed across all tasks, highlighting the utility of the structured latent space to behavior models.

**Discovering Task-relevant Objects**  In Figure 6, we visualize an extract of the slot history via the image reconstructions visualized in the first row. To visualize the attention pattern of the actor in the current (rightmost) time-step we multiply the attention scores with the masks of the respective objects and show them overlaid with the RGB reconstructions and as an individual colormap in the second and third row respectively. This visualization shows the *Push-Specific* task and we find that the model discovers task relevant objects automatically, ignoring information stored in slots that represent distractor objects across all time-steps, while the robot and green cube receive the most attention. While the recency bias induced by ALiBi is clearly visible, we find that the model learns to pierce through it when necessary, attending to the target, which has been occluded for 15 time-steps in the last time-step where it was visible. We see that the model is able to focus on task-relevant information effectively, even when reasoning over long time sequences is required.

**SAVi Finetuning**  One common limitation of prior works is that object-centric models are pre-trained on sequences with random behaviors but then frozen during training. This restricts their applicability to tasks where the state distributions encountered by random and successful policies are similar. With the *PickAndPlace* tasks, we explicitly violate this assumption since random be-

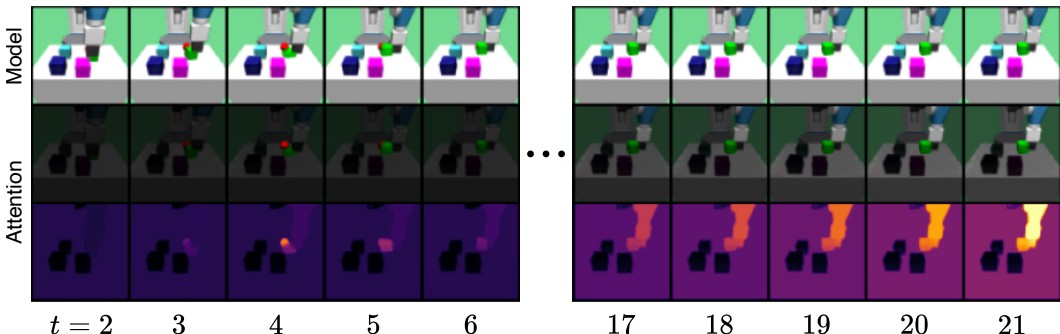

Figure 6: SOLD discovers objects relevant for task completion in an unsupervised manner over long horizons. We depict the normalized attention of the [out] token of the actor over the object tokens using Attention Rollout (Abnar & Zuidema, 2020). The full slot history is shown in Figure 15.

haviors are highly unlikely to pick and lift blocks off the table. Hence, SAVi has not seen blocks in the air, a configuration that will necessarily be reached by performant policies. Figure 7 illustrates the need to continually fine-tune the object-centric encoder-decoder model. While the fine-tuned SAVi model is able to reconstruct the lifted block accurately, the frozen variant fails to decode this configuration correctly. The target block effectively dissolves when lifted, hindering the discovery of behaviors associated with such states.

**Generalization to Non-Object-Centric Environments** While it is commonplace to evaluate object-centric methods on environments and datasets that naturally lend themselves to such decompositions, we aim to showcase the potential of our methods to generalize beyond this setting. To this end, we tested our method on Meta-World to assess its performance on tasks involving objects with complex shapes and colors, rather than simple uni-colored objects. Our method converges to success rate of $100\%$ on both the *Button-Press* and *Hammer* task. Further, we test on DM-Control to evaluate generalization to problems that are vastly different and where reasoning over objects and interactions is less pronounced. On the *Cartpole-Balance* and *Finger-Spin* tasks, we reach returns of $497$ and $645$, respectively. We provide details about the environment decomposition and dynamics prediction for all four tasks in Section F in the Appendix.

## 5 RELATED WORK

**Object-Centric Learning** In recent years, the field of unsupervised object-centric representation learning from images and videos has gained significant attention (Yuan et al., 2023). Most existing methods follow an encoder-decoder framework with a structured bottleneck composed of $N$ latent vectors called slots, where each of these slots binds to a different object in the input image. Slot-based methods have been widely applied for images (Burgess et al., 2019; Engelcke et al., 2020; Locatello et al., 2020; Singh et al., 2021; Engelcke et al., 2021; Singh et al., 2023; Biza et al., 2023) and videos (Kipf et al., 2022; Singh et al., 2022; Elsayed et al., 2022; Bao et al., 2022; Zoran et al., 2021; Kabra et al., 2021; Creswell et al., 2021). However, despite their impressive performance on synthetic datasets, they often fail to generalize to visually complex scenes. To overcome this limitation, recent methods propose introducing some weak supervision (Elsayed et al., 2022; Bao et al., 2023), levering large self-supervised pretrained encoders (Seitzer et al., 2023; Zadaianchuk et al., 2024; Aydemir et al., 2023; Kakogeorgiou et al., 2024), or using diffusion models as slot decoders (Jiang et al., 2023; Wu et al., 2023b; Singh et al., 2024).

**Object-Centric Video Prediction** Object-centric video prediction aims to understand the object dynamics in a video sequence with the goal of anticipating how these objects will move and interact with each other in future time steps. With this end, multiple methods propose to model and forecast the object dynamics using different architectures, including RNNs (Zoran et al., 2021; Nakano et al., 2023) transformers (Wu et al., 2023a; Villar-Corrales et al., 2023; Song et al., 2023; Gandhi et al., 2024; Daniel & Tamar, 2024; Nguyen et al., 2024; Petri et al., 2024) or state-space models (Jiang et al., 2024), achieving an impressive prediction performance on synthetic video datasets

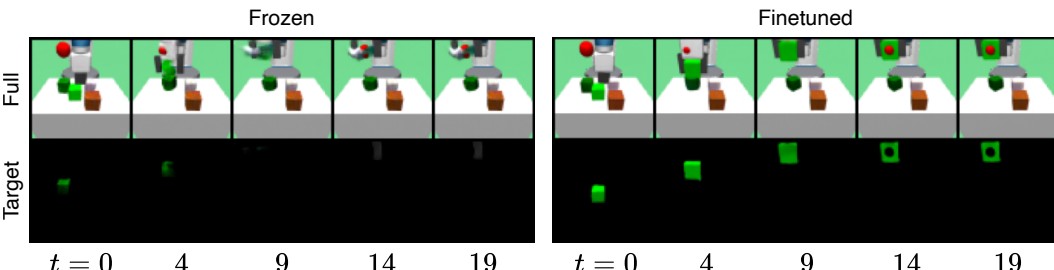

Figure 7: Visualization of the full reconstruction and the slot that represents the target object for a frozen and finetuned SAVi model.

and learning representations that can help solve downstream tasks that require reasoning about objects properties and relationships (Wu et al., 2023a; Petri et al., 2024).

**Model-based Reinforcement Learning**  Model-based reinforcement learning approaches aim to improve sample efficiency by learning environment dynamics. PlaNet (Hafner et al., 2019) introduced a latent dynamics model for efficient planning, while the Dreamer family (Hafner et al., 2020; 2021; 2023) incorporated this into an actor-critic framework. DreamerV2 and DreamerV3 introduced further improvements like categorical latent states and robustness techniques. DreamerV3 has shown superior performance in visual control tasks compared to model-free approaches, but uses holistic rather than object-centric state representations.

**Reinforcement Learning with Object-Centric Representations**  Recent works have explored integrating object-centric representations into RL frameworks. SMORL (Zadaianchuk et al., 2021) and EIT (Haramati et al., 2024) combined object-centric representations with goal-conditioned model-free RL for robotic manipulation. Yoon et al. (2023) investigated pre-training object-centric representations for RL, showing benefits for relational reasoning tasks. The field of object-centric model-based RL is still largely underexplored. One approach that can be categorized as such is FOCUS (Ferraro et al., 2023). However, unlike our method, FOCUS does not use the object-centric states in forward prediction or action selection, but mainly for an exploration target. Further, FOCUS requires supervision via ground-truth segmentation masks to learn the object-centric states.

## 6    CONCLUSION

We present SOLD, an object-centric model-based RL algorithm that learns directly from pixel inputs. By employing structured latent representations through slot-based dynamics models, our method offers a compelling alternative to traditional, holistic approaches. While object-centric representations have been valued for their role in forward prediction (Villar-Corrales et al., 2023), we demonstrate their synergistic benefits in accelerating the learning of behavior models. SOLD achieved strong performance across visual robotics environments, significantly outperforming the state-of-the-art DreamerV3, particularly in tasks requiring relational reasoning. Additionally, the learned behavior models exhibit interpretable attention patterns, explicitly focusing on task-relevant parts of the visual scene.

**Limitations & Future Work**  One limitation of our world model is that it generates predictions in a deterministic manner. This can be a drawback in environments that are inherently stochastic or highly unpredictable. We believe this is a key reason why SOLD outperforms DreamerV3 on complex robotic manipulation tasks but struggles to match its performance on simpler tasks like *Cartpole-Balance*, where minor variations in action sequences can lead to vastly different outcomes over long horizons. Addressing this limitation by incorporating stochasticity into the prediction model presents a promising direction for future work. A second limitation arises from the object-centric encoder-decoder model we use. While SAVi performs well on the tasks we evaluated, scaling it to complex real-world data remains a significant challenge. However, the core ideas of our method are independent of the specific object-centric encoder-decoder model, and future work can easily integrate more advanced models that address these scalability concerns.

## REPRODUCIBILITY STATEMENT

With the goal of reproducible research, we conducted three runs per method and task, each with different random seeds to account for seed-dependent variability. Additionally, we thoroughly list in Appendix C the model architectures and hyper-parameters used in our experiments. Upon acceptance of the paper, we will open-source our code, environments and pretrained models.

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

## A    NOTATION

**Slot Attention for Video (SAVi)**

| | |
|---|---|
| $D_Z$ | The slot dimension |
| $N$ | The number of slots |
| $\boldsymbol{Z}_t$ | A set of slots $\boldsymbol{Z}_t \in \mathbb{R}^{N \times D_Z}$ at time-step $t$ |
| $\boldsymbol{Z}_{0:t}$ | A history of slot-sets up to time-step $t$ |
| $e_\psi$ | A SAVi encoder that maps $\boldsymbol{o}_t$ to $\boldsymbol{Z}_t$ |
| $d_\psi$ | A SAVi decoder that reconstructs $\boldsymbol{o}_t$ from $\boldsymbol{Z}_t$ |
| $\boldsymbol{F}_t$ | Features obtained by encoding images |
| $L$ | Number of spatial locations in $\boldsymbol{F}$ |

**Reinforcement Learning**

| | |
|---|---|
| $S$ | The number of seed frames |
| $T$ | The imagination horizon |
| $\boldsymbol{o}_t$ | An image observation |
| $\boldsymbol{a}_t$ | An action command |
| $r_t$ | A reward |
| $\gamma$ | A scalar discount factor |
| H | The entropy of a probability distribution |
| $f_\alpha^{\text{MLP}}$ | An MLP head that belongs to parameter group $\alpha$ |
| $\boldsymbol{h}_t$ | A processed output token |

## B    EXTENDED RELATED WORK

**Model-based Reinforcement Learning**    Model-based methods hold the potential to significantly improve the sample efficiency of RL algorithms, and recent years have seen several key contributions advancing this area.

Pioneering work by Ha & Schmidhuber (2018) introduced the concept of a recurrent generative model, termed a world model, which captures the dynamics of visual RL environments. By encoding high-dimensional observations into a compact latent representation, this model enables RL agents to train policies entirely within imagined rollouts.

The Planning Network (PlaNet) (Hafner et al., 2019) introduced a recurrent state-space model (RSSM) that predicts future states directly in a compact latent space, avoiding the costly step of decoding full observations. This architecture enables efficient planning of action sequences but is limited by short planning horizons. Building on this, Dreamer (Hafner et al., 2020) integrates planning and learning by training agents within a learned world model, overcoming PlaNet's short-sighted horizon. Subsequent versions, DreamerV2 (Hafner et al., 2021) and DreamerV3 (Hafner et al., 2023), improved robustness and generalization through enhanced representation learning and optimization techniques, achieving state-of-the-art results across diverse RL benchmarks.

Temporal Difference Learning for Model Predictive Control (TD-MPC) (Hansen et al., 2022) introduced a task-oriented latent dynamics model to optimize trajectories directly within the latent space of a world model. Unlike earlier approaches, TD-MPC avoids reconstructing full observations, instead focusing the world model on reward-predictive elements through a loss applied to reward and value predictions. TD-MPC2 (Hansen et al., 2024) builds on this by introducing scalability improvements, enabling superior performance with larger model sizes and demonstrating a single agent's ability to generalize across multiple tasks and action spaces.

Table 1: Implementation details for each of SOLD modules.

| (a) SAVi | | (b) Object-centric dynamics | | (c) Slot Aggregation Transformer | |
|---|---|---|---|---|---|
| **Hyper-Param.** | **Value** | **Hyper-Param.** | **Value** | **Hyper-Param.** | **Value** |
| Slot Dim. $D_Z$ | 128 | # Layers | 4 | # Layers | 4 |
| # Slots $N$ | 2-10 | # Heads | 8 | # Heads | 8 |
| Slot Init. | Learned | Token Dim. | 256 | Token Dim. | 256 |
| # Iters. | 3/1 | MLP Dim. | 512 | MLP Dim. | 512 |

Inspired by the success of Transformers on sequence modeling tasks, Micheli et al. (2023) proposed IRIS, a method combining a discrete autoencoder with an autoregressive Transformer to model environment dynamics. The autoencoder tokenizes images into a discrete set of representations, while the Transformer learns temporal dynamics across these tokens. IRIS demonstrated visually and temporally accurate predictions of game dynamics in Atari environments. While IRIS shares similarities with our approach – encoding an image into a set-based representation and predicting it forward using a Transformer – it lacks the object-centric interpretability afforded by our model.

Recently, Alonso et al. (2024) proposed diffusion as a promising alternative to discretization of the latent space of the world model. DIAMOND uses a diffusion-based conditional generative model, $p(\boldsymbol{x}_{t+1} \mid \boldsymbol{x}_{\leq t}, \boldsymbol{a}_{\leq t})$, to produce visually precise next-frame predictions. The authors demonstrate the potential of their world model to simulate complex 3D environments by learning a realistic game-engine from static Counter-Strike: Global Offensive gameplay. Valevski et al. (2024) also propose to use diffusion to create high-quality visual predictions. Specifically, they demonstrate the potential of diffusion models to serve as real-time game engines.

## C    IMPLEMENTATION DETAILS

In this section, we describe the network architecture and training details for each of the SOLD components. Our models are implemented in PyTorch (Paszke et al., 2019), have 12 million learnable parameters, and are trained on a single NVIDIA A-100 GPU with 40GB of VRAM. A summary of the model implementation details is listed in Table 1.

### C.1    SLOT ATTENTION FOR VIDEO

We closely follow Kipf et al. (2022) for the implementation of the Slot Attention for Video (SAVi) decomposition model, including their proposed CNN-based encoder $e_\psi$ and decoder $d_\psi$, the transformer-based predictor and the Slot Attention corrector. We employ between 2 and 10 (depending on the environment) 128-dimensional object slots, whose initialization is learned via backpropagation. We empirically verified that learning the initial slots performs more stable than the usual random initialization. Furthermore, we use three Slot Attention iterations for the first video frame in order to obtain a good initial decomposition, and a single iteration for subsequent frames, which is enough to update the slot state given the observed features.

### C.2    OBJECT-CENTRIC DYNAMICS MODEL

Our object-centric dynamics model is based on the OCVP-Seq (Villar-Corrales et al., 2023) architecture, which is a transformer encoder employing sequential and relational attention mechanisms in order to decouple the processing of temporal dynamics and interactions, and has been shown to achieve interpretable and temporally consistent predictions. We use 4 transformer layers employing 256-dimensional tokens, 8 attention heads, and using a hidden dimension of 512 in the feed-forward layers.

### C.3    SLOT AGGREGATION TRANSFORMER

The *Slot Aggregation Transformer* (SAT) forms the architectural backbone for the reward, value and action models. This module aggregates information from object slots across multiple time steps to

Figure 8: The *Slot Aggregation Transformer* applies causal masking and ensures that output and register token do not attend to themselves on other time-steps. The recency bias induced by ALiBi is visualized through the color gradient in the attention mask, with lighter shades of blue corresponding to a higher negative bias on the attention scores.

produce output tokens that are subsequently fed to MLP heads in order to predict rewards, values, or actions. An overview of our SAT module is depicted in Figure 8.

SAT is a causal transformer encoder module that receives as input a history of object slots, as well as a learnable output token `[out]` for each time step, which is responsible for producing the final output for the corresponding time step. Additionally, we append to the SAT inputs a number of register tokens `[reg]` per time-step, which have been shown to aid with processing in attention-based models by offloading intermediate computations from the output tokens and helping the module focus on relevant slots (Darcet et al., 2024).

To encode the positional information into SAT, we employ *Attention with Linear Biases* (Press et al., 2022) (ALiBi), which introduces linear biases directly into the attention scores, effectively encoding token recency. This approach helps the model deal with sequences of varying length, as well as generalize to longer sequences than those seen during training, thus outperforming absolute positional encodings.

For our experiments, SAT is implemented with 4 transformer encoder layers with causal self-attention, RMS-Normalization layers (Zhang & Sennrich, 2019), 8 attention heads, a token dimension of 256, and a hidden dimensionality in the feed-forward layers of 512. We set the number of learnable register token per time step to 4. Furthermore, we enforce in our causal attention masks that tokens belonging to time step $t$ cannot directly interact with previous output and register tokens.

## C.4 TRAINING DETAILS

**SAVi Pretraining** SAVi is pretrained for object-centric decomposition on approximately one million frames for 400,000 gradient steps. We use the Adam optimizer (Kingma & Ba, 2015), a batch size of 64 and a base learning rate of $10^{-4}$, which is first linearly warmed-up during the first 2,500 training steps, followed by cosine annealing (Loshchilov & Hutter, 2017) for the remaining of the training procedure. We perform gradient clipping with a maximum norm of 0.05.

**SOLD Training** SOLD is trained using the Adam optimizer (Kingma & Ba, 2015) and different learning rates for each component: $10^{-4}$ for the dynamics and rewards models, and $3 \cdot 10^{-5}$ for training the action and value models, as well as for fine-tuning the SAVi encoder. To stabilize training, we perform gradient clipping with maximum norm of 0.05 for the SAVi model, 3.0 for the transition model, and 10.0 for the reward, value, and action models. For all components, we also use learning rate warmup for the first 2,500 gradient steps. Additionally, we implement the exponential moving average (EMA) for the target value network with a decay rate of 0.98. We use an imagination horizon of 15 steps for behavior learning, and the $\lambda$-parameter is set to 0.95.

## D  BASELINES

In our experiments we compare our approach with two different baseline models, namely the SoTA model-based RL baseline *DreamerV3* and a *Non-Object-Centric* variant of our proposed model:

**DreamerV3** DreamerV3 (Hafner et al., 2023) is a SoTA model-based reinforcement learning algorithm that learns behaviors from visual inputs without requiring task-specific inductive biases or extensive environment interaction. It builds a world model that predicts future states and rewards, which is then used to simulate potential outcomes and guide the agent's decision. DreamerV3 leverages latent dynamics and a compact, holistic representation of the environment for an efficient exploration, while showing desirable properties such as sample efficiency, scalability, and generalization across a wide range of complex tasks. We select the 12 million parameter variant so as to match the parameter count of our proposed model. For further details, we refer to Hafner et al. (2023).

**Non-Object-Centric Baseline** This baseline model follows the same general framework as our proposed model, but replaces the object-centric SAVi encoder and decoder with a simple convolutional auto-encoder while keeping the remaining modules unchanged; thus allowing us to ablate the effect of object-centric representations for model-based reinforcement learning. The CNN auto-encoder used in this baseline consists of an encoder comprised of four strided convolutional layers with 64, 128, 256, and 512 channels respectively, each followed by batch normalization and a ReLU. The output of the final convolutional layer is flattened and fed through a linear layer to produce a 512-dimensional latent vector. The decoder mirrors the encoder structure, reconstructing the observations from the latent representation through the use of four transposed convolutional layers. To compensate for the lack of multiple latent vectors and to ensure a fair comparison, we increase the capacity of this baseline model by scaling the actor, critic, and dynamics models. Specifically, we increase the token dimension from 256 to 512, as well as the MLP hidden dimension from 512 to 1024. The total parameter count for this baseline is approximately 60 million, thus being five times larger than our proposed method and the DreamerV3 baseline.

## E  ENVIRONMENTS

In this section, we provide further details about our proposed suite of environments, which includes eight object-centric robotic control tasks designed to test relational reasoning and manipulation capabilities. The environments, which are inspired by Li et al. (2020) and are simulated using Mu-JoCo (Todorov et al., 2012), follow the same basic structure, consisting of a robot arm mounted on a base, positioned near a table where the manipulation tasks take place.

In all environments, the robot is controlled by a 4-dimensional action vector $\boldsymbol{a} = [a_x, a_y, a_z, a_{grip}] \in [-1, 1]^4$, where the first three components represent the desired movement direction of the end-effector, whereas the fourth component controls the gripper. On the *Reach* and *Push* tasks, commands to the gripper are ignored, with the gripper fixed in a closed configuration, as gripping is not required to solve these tasks.

For all tasks, we define the following constants:

- $t_1 = 20$ and $t_2 = 10$: Temperature parameters that determine the steepness of the reward function.
- $d_m = 0.05$: Distance threshold (in meters) for considering a task successful.

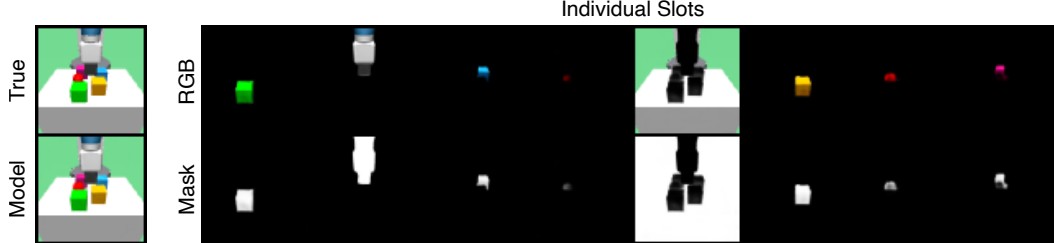

Figure 9: Object-centric SAVi decomposition of a video frame. We show the masked RGB image and the segmentation mask corresponding to each object slot. The masked RGB images are combined to reconstruct the observed frame.

**Reach**  In *Reach* tasks, the agent must identify a spherical target among several distractors and mode the end-effector to its location. The reward is calculated as:

$$r = \exp(-t_1 \cdot ||\boldsymbol{p}_e - \boldsymbol{p}_t||_2), \tag{12}$$

where $\boldsymbol{p}_e$ is the position of the end-effector and $\boldsymbol{p}_t$ is the target position. The success at the end of an episode is defined as:

$$\text{success} = \begin{cases} 1 & \text{if } ||\boldsymbol{p}_e - \boldsymbol{p}_t||_2 < d_m \\ 0 & \text{otherwise} \end{cases}. \tag{13}$$

**Push & PickAndPlace**  Both *Push* and *PickAndPlace* correspond to reasoning and manipulation tasks where the agent must identify a single block among several distractors and move it to a target location. In *Push* tasks, the agent can slide the block to the target position on the table without using its gripper; whereas in *PickAndPlace* the target location can be above the table, thus needing to grasp the block in order to lift it to the target position. In both task variants the reward is calculated as:

$$r = 0.9 \cdot \exp(-t_1 \cdot ||\boldsymbol{p}_c - \boldsymbol{p}_t||_2) + 0.1 \cdot \exp(-t_2 \cdot ||\boldsymbol{p}_e - \boldsymbol{p}_c||_2), \tag{14}$$

where $\boldsymbol{p}_e$ is the position of the end-effector, $\boldsymbol{p}_t$ is the target position, and $\boldsymbol{p}_c$ is the block position. The success at the end of an episode is defined as:

$$\text{success} = \begin{cases} 1 & \text{if } ||\boldsymbol{p}_c - \boldsymbol{p}_t||_2 < d_m \\ 0 & \text{otherwise} \end{cases}. \tag{15}$$

# F  ADDITIONAL RESULTS

## F.1  OBJECT-CENTRIC DECOMPOSITION

Figure 9 depicts the object-centric decomposition of a video frame obtained by SAVi. SAVi parses the input frame into per-object RGB reconstructions and alpha masks, which can be combined via a weighted sum in order to accurately reconstruct the observed video frame. Notably, SAVi assigns an object slot to the scene background, five slots to different blocks, one slot to the red target, and one slot to the robot arm. The sharp object masks demonstrate that SAVi isolates object-specific information in each slot, which is beneficial for downstream applications such as behavior learning, allowing the agent to reason about object properties and their relationships while abstracting task-irrelevant details.

## F.2  OPEN-LOOP PREDICTION

We visualize action-conditional open-loop predictions on the *Push Specific* (Figure 10), *Button-Press* (Figure 11), *Hammer* (Figure 12), *Cartpole-Balance* (Figure 13), and *Finger-Spin* (Figure 14) environments. More precisely, we depict the ground truth sequence, the predicted video frames, the predicted instance segmentation of the scene, obtained by assigning a distinct color to each object mask, and the object reconstruction for each slot.

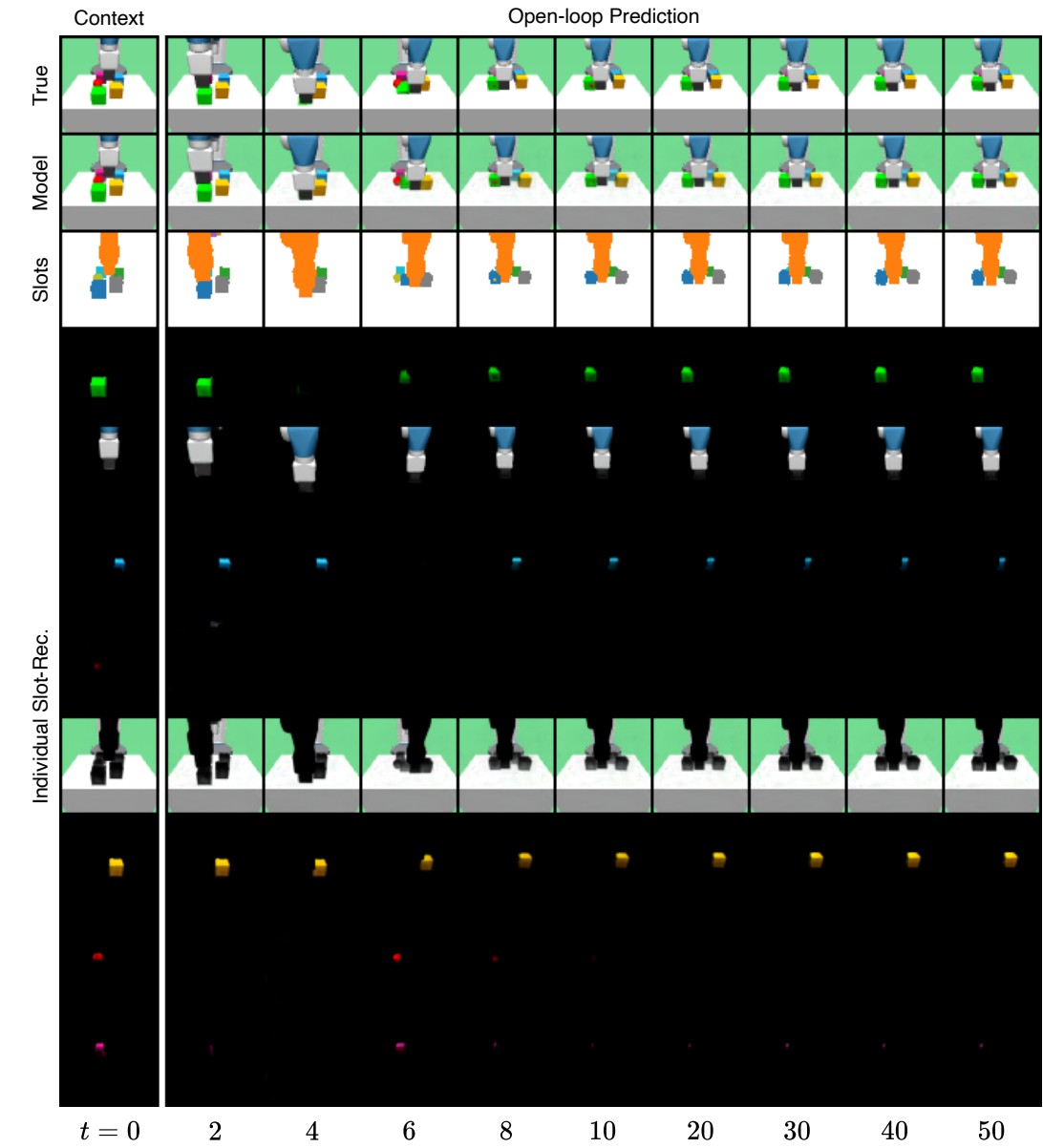

Figure 10: Open-loop prediction on the *Push-Specific* task. We visualize the ground truth, predicted frames, segmentation obtained by assigning different colors to each object mask, and per-object reconstructions. In this sequence, SOLD assigns one slot to the background, one slot for the robot, one slot for the target, and four different slots for blocks, while one slot remains empty.

In all examples, our model parses the scene into sharp and accurate object representations, and models the action-conditional object dynamics and interactions in order to accurately predict the future video frames while preserving object-centric representations. We emphasize SOLD's ability to capture complex physical interactions, such as pushing a block to a target location (Fig. 10), pressing a button (Fig. 11) or hitting a nail with a hammer (Fig. 12). Furthermore, we showcase that SOLD can generalize to diverse non-object-centric environments (Fig. 13 and Fig. 14).

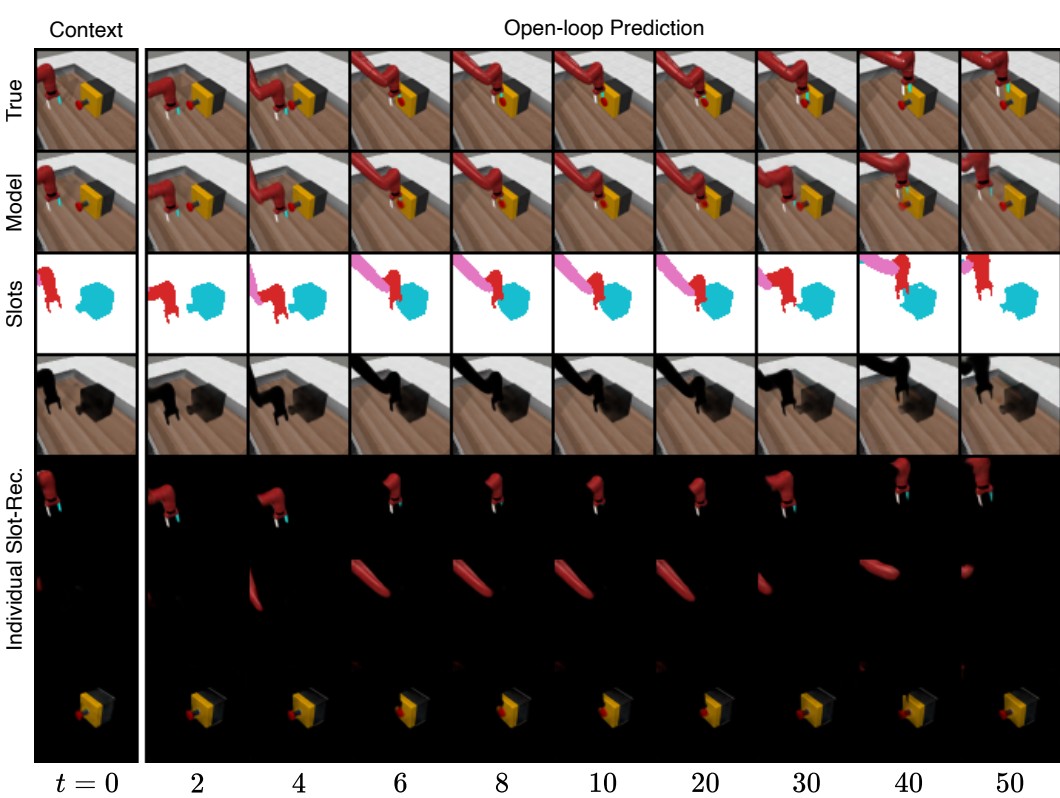

Figure 11: Open-loop prediction and decomposition results on the *Button-Press* task. We visualize the ground truth and predicted video frames, instance segmentation obtained by assigning a different color to each object mask, and per-object reconstructions. SOLD assigns a slot for the scene background, two slots for different robot parts, and a slot for the button-box.

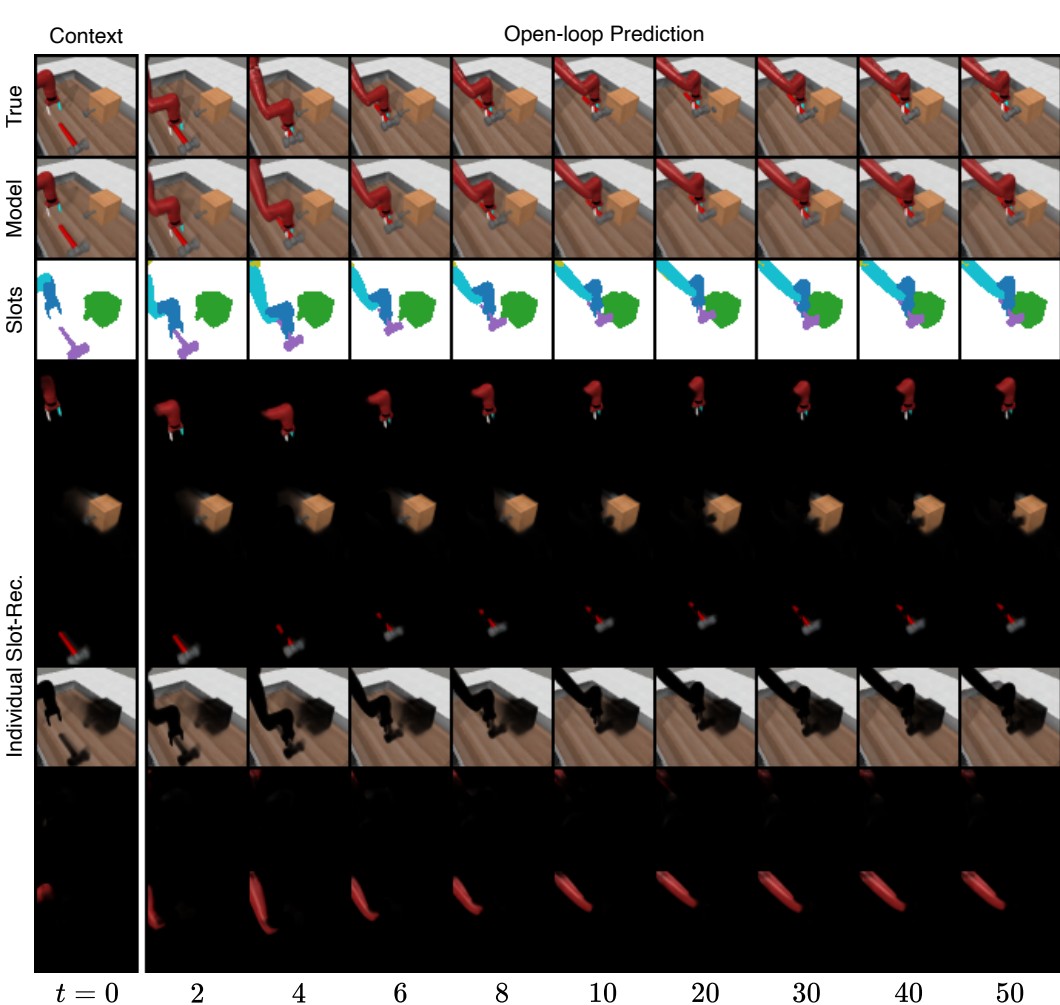

Figure 12: Open-loop prediction and decomposition results on the *Hammer* task. We visualize the ground truth and predicted video frames, instance segmentation obtained by assigning a different color to each object mask, and per-object reconstructions. SOLD assigns a slot for the scene background, three slots for different robot parts, a slot for the hammer, and a slot for the nail-box.

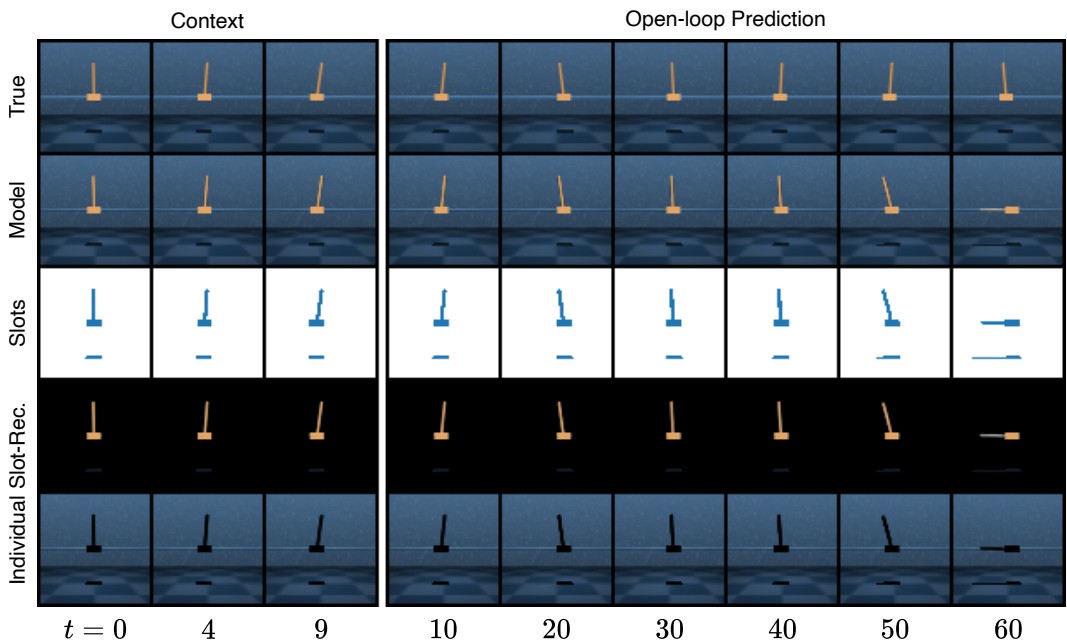

Figure 13: Open-loop prediction and decomposition results on the *Cartpole-Balance* task. We visualize the ground truth and predicted video frames, instance segmentation obtained by assigning a different color to each object mask, and per-object reconstructions. SOLD assigns a slot for the scene background and a slot for the cart-pole. Notably, the slot represents the object along with its shadow.

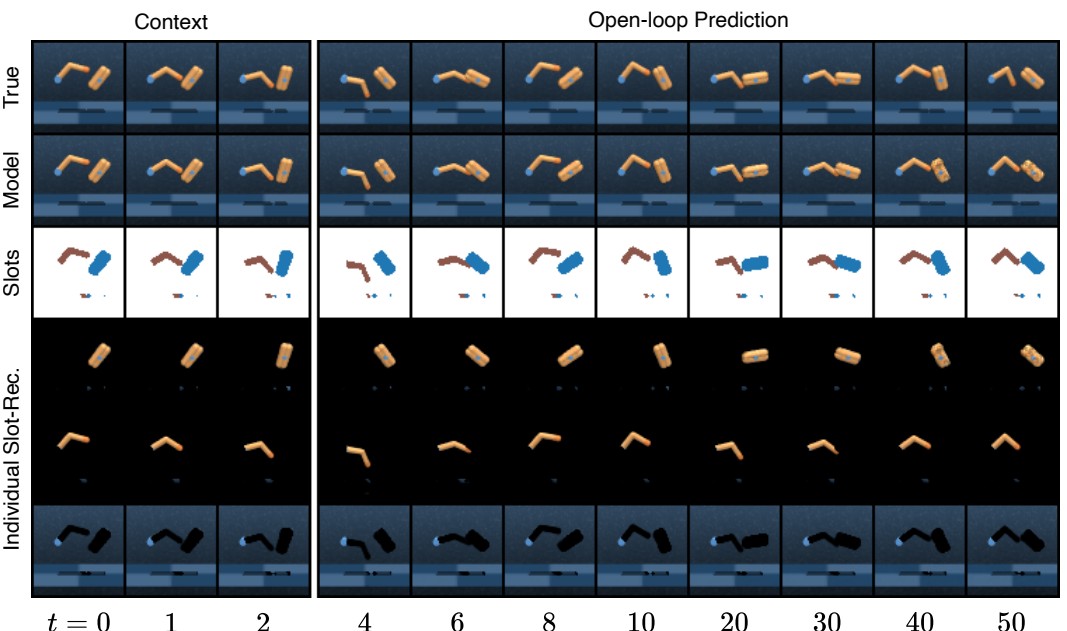

Figure 14: Open-loop prediction and decomposition results on the *Finger-Spin* task. We visualize the ground truth and predicted video frames, instance segmentation obtained by assigning a different color to each object mask, and per-object reconstructions. SOLD assigns a slot for the scene background, a slot for the finger, and a slot for the spinning target. Notably, the slots represent the objects along with their corresponding shadows.

1296
1297
1298
1299
1300
1301
1302
1303
1304
1305
1306
1307
1308
1309
1310
1311
1312
1313
1314
1315
1316
1317
1318
1319
1320
1321
1322
1323
1324
1325
1326
1327
1328

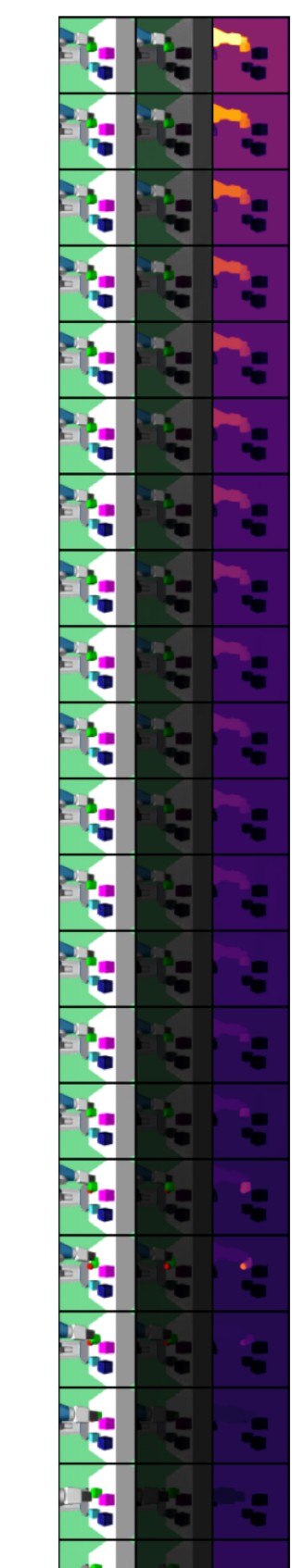

Figure 15: Original, uncut slot-history used in Figure 6. The full rollout highlights both the recency bias introduced by ALiBi and the model's ability to overcome this bias when task-relevant information must be retrieved from slots far in the past.

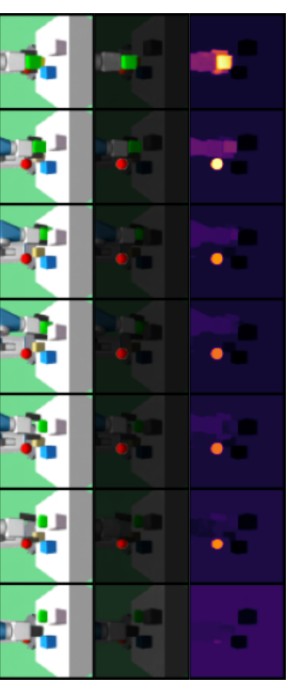

Figure 16: Visualizing the actor's attention over the slot-history on the *PickAndPlace-Specific* task reveals that the robot's gripper and target block in the current time-step receive the most attention, highlighting the model's ability to focus on task-relevant components. Further, the red target sphere is mostly occluded in the current time-step making it difficult to reconstruct its position accurately. However, the model learns to integrate information from previous steps, where it was still clearly visible.

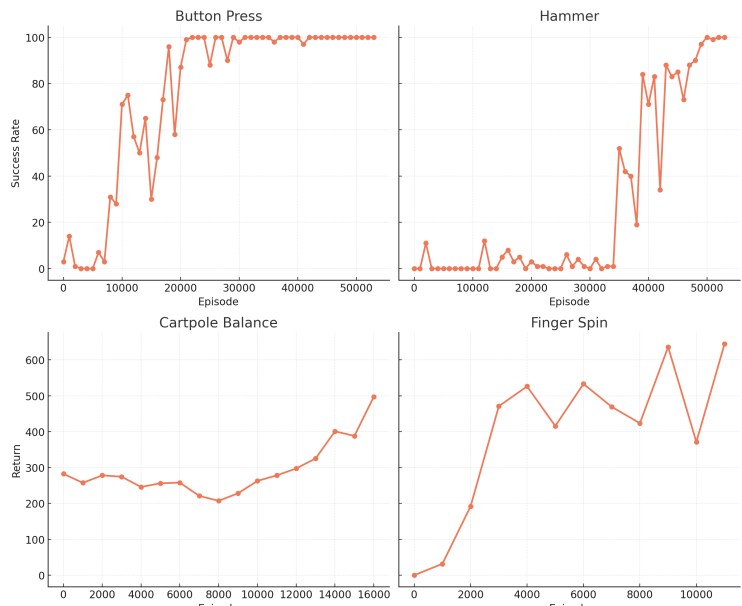

Figure 17: Training curves for the non-object-centric environments studied in this work.

## F.3 QUANTITATIVE EVALUATION

We present the training performance of SOLD on non-object-centric environments for a single seed in Figure 17. Although SOLD is not expected to outperform baseline methods in environments that do not rely on relational reasoning or significantly benefit from object-centric decomposition, our results highlight the potential of applying object-centric representations to various visual domains. This is underscored by SOLD's ability to generalize to the complex hammer and button press tasks, converging to a success rate of 100% on both problems.

## F.4 GENERALIZATION TO DIFFERENT COLORS

To investigate the generalization beyond the training distribution, we evaluate the performance of all methods on a different color set for all tasks. The results are shown in Figure 18. When comparing to the original evaluation in Figure 4 we observe a very small drop in performance for both SOLD and DreamerV3, indicating robustness to changes in the object color.

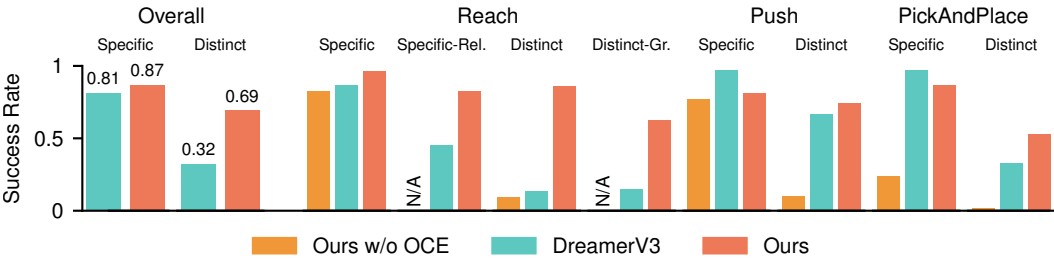

Figure 18: Success rates after changing the color set to 32 novel colors.

## F.5 SAVi FINE-TUNING

Figure 19 illustrates the importance of fine-tuning the object-centric encoder-decoder model with another example. Without fine-tuning, the blue color, which appears similarly on both colored blocks and the robot arm, leads to an even more drastic degradation of the reconstructions, where the robot

True Observations

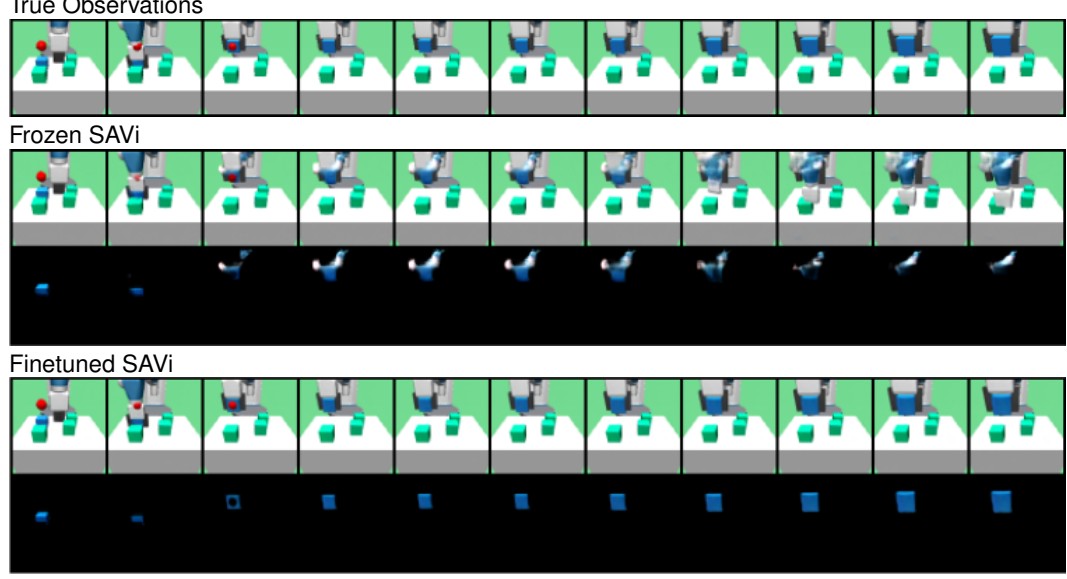

Frozen SAVi

Finetuned SAVi

Figure 19: Comparison of fine-tuned and frozen SAVi models on PickAndPlace-Distinct. We visualize the full reconstruction and the slot that reconstructs the cube that is being lifted for both models. When the blue block is lifted off the table, the frozen model merges it with blue elements from the robot arm, deteriorating the prediction and hallucinating the arm going between the gripper fingers. The fine-tuned model, on the other hand, is able to reconstruct the sequence accurately.

itself is no longer accurately captured. In contrast, the fine-tuned model is able to reconstruct the sequence accurately.

