# OpenReview forum: "Learning Object-centric Latent Dynamics for Reinforcement Learning from Pixels"
_ICLR.cc/2025/Conference — Submitted to ICLR 2025_

### Official Review · Reviewer_W43M · 2024-10-30

**Soundness:** 3
**Presentation:** 3
**Contribution:** 2
**Rating:** 3
**Confidence:** 5

**Summary:**

This paper introduces the use of slot attention to learn object-centric latent dynamics and employs a slot-aggregated transformer as the backbone for a reward/actor/critic model. Techniques such as register tokens and ALiBi are incorporated to enhance the performance of the model-based RL algorithm, DreamerV3. The central premise is to develop structured latent world models that differentiate various objects in a scene, thereby aiding behavior learning. Experiments across 12 simulation tasks demonstrate significant performance improvements over the original DreamerV3.

**Strengths:**

- The paper is well-written and organized, featuring numerous visualizations that facilitate understanding of the proposed object-centric world models.
- Experimental results indicate notable performance gains over the state-of-the-art DreamerV3 across a diverse array of robotics tasks (e.g., DM-Control, Meta-World).
- The concept of utilizing slot attention to learn structured world models is both intuitive and effectively illustrated, though I have some questions about it.

**Weaknesses:**

One major concern is the justification for the necessity of object-centric latent dynamics in model-based RL:

- The objectives of employing latent dynamics include (1) reducing rollout time and (2) maintaining accurate world modeling. However, the paper lacks quantitative comparisons regarding time efficiency and accuracy against DreamerV3.
- The ability of the world model to handle more complex scenes remains unclear. The current tasks appear too simplistic; it is recommended to utilize more realistic simulators and diverse object tasks to assess whether the proposed method surpasses earlier models like DreamerV3.
- The discussion on the additional properties of the proposed world model is insufficient. For instance, will the learned model demonstrate greater robustness to scene disturbances, such as sudden changes in background, object textures, or the number of objects? Additionally, will it model interactions between objects and manipulators more accurately?
- The benefits for behavior learning are not clear as well. While performance improvements in simulation tasks are shown, more insights into how object-centric dynamics contribute to this enhancement—such as increased accuracy of value estimation by critic—are suggested.

My second concern involves the lack of discussion and comparison with state-of-the-art model-based RL algorithms:

- There is no mention, comparison, or citation of other SOTA algorithms, such as TD-MPC(2) [1], which is strange for a model-based RL paper.
- The applicability of the proposed world model to other methods like TD-MPC(2) is unclear, raising questions about the potential for extending the proposed models to other MBRL methods or surpassing them.

The experiments could be more comprehensive.

- An ablation study on key design choices is missing, such as slot attention, register tokens, ALiBi, and the slot-aggregated transformer for reward/actor/critic components.
- The experiments are limited to gripper-based manipulation tasks and simple manipulation skills. Investigating performance on more complex dexterous-hand tasks [2] or humanoid tasks would be beneficial.
- The effectiveness of the proposed method on object-free tasks, such as locomotion tasks in DM-Control, is unclear.

**Questions:**

- What distinguishes "Ours w/o OCE" from DreamerV3? If the differences stem solely from the proposed tricks, does this imply that these tricks may be detrimental? This raises concerns about the practical implementation of all methods.
- Are tricks like EMA, spatial bins, and imagination also utilized in DreamerV3? This raises issues regarding the fairness of comparisons and, consequently, the true benefits of the proposed components.
- What is the effect of removing pre-training stage of world model? What’s the performance of DreamerV3 if pre-training its world model instead of learning from scratch?
- In Figure 6, the authors claim that SOLD discovers objects relevant to tasks. However, at the initial timesteps (e.g., t=2), the object is not discovered until the manipulator begins to interact with it (after t=3). This raises the chicken-and-egg dilemma: does object discovery drive manipulation, or vice versa? Could the authors clarify this?
- Still, regarding object discovery, can DreamerV3 or SOLD w/ CNN already discover target objects? Visualizing their attention maps using techniques like Grad-CAM could help address this concern.
- It is good to visualize slot representations in Figure 3. But it seems that SAVi could achieve similar results using videos alone, without action/reward labels. How do action/reward labels influence the representational capacity of SAVi in this context?

[1] Hansen, N., Su, H., & Wang, X. TD-MPC2: Scalable, Robust World Models for Continuous Control. In *The Twelfth International Conference on Learning Representations*.

[2] Sferrazza, C., Huang, D. M., Lin, X., Lee, Y., & Abbeel, P. (2024). Humanoidbench: Simulated humanoid benchmark for whole-body locomotion and manipulation. *arXiv preprint arXiv:2403.10506*.

---

> ### Author Response · Authors · 2024-11-19
> **Response to Reviewer W43M (1/2)**
>
> We thank the reviewer for the constructive review!
> Following your advice we added new sections to the appendix including the following:
> - review of further Model-Based RL related works (App. B),
> - training curves depicting the success-rate and return of our proposed SOLD model on the Meta-World and DM-Control environments, respectively (App. F.3), and
> - an ablation study evaluating the importance of SAVi fine-tuning during RL training (App. F.5).
>
> We are currently extending the ablation study with further quantitative evaluations and measuring the generalization capabilities of our model. Below we address highlighted weaknesses and questions. To address everything in detail, we break down our reply into two comments.
>
> **The effectiveness of the proposed method on object-free tasks, such as locomotion tasks in DM-Control, is unclear.**
>
> SOLD is specifically designed to excel in environments where object-centric decompositions and relational reasoning between objects play a crucial role, as is often the case in robotic manipulation tasks. To demonstrate the versatility of our approach, we also evaluate SOLD on non-object-centric environments as a proof-of-concept, showcasing its ability to generalize to other task domains. For example, in Cartpole-Balance, the object-centric decomposition isolates only two slots: one for the robot (cart and pole) and one for the background. In such scenarios, akin to locomotion tasks in DM-Controls, the inductive bias of decomposing the scene into objects provides limited advantage, and SOLD is not expected to outperform baseline methods. Nonetheless, we demonstrate that SOLD remains applicable and that reasonable behavior is learned, even in these general visual RL settings where object-centric reasoning is less beneficial.
>
> **What distinguishes "Ours w/o OCE" from DreamerV3?**
>
> Our SOLD w/o OCE corresponds to a variant of our proposed SOLD model, but replacing our structured state composed of multiple object slots, with a single holistic embedding representation.
> Therefore, some architectural components in SOLD w/o OCE and DreamerV3 present fundamental differences, whereas the ‘tricks’ used for behavior learning are present in both SOLD w/o OCE and DreamerV3:
> Predictor: DreamerV3 uses a stochastic recurrent RSSM world model implemented with Block-GRU modules, which receive as input at every time-step an embedding, the action vector and the previous GRU state. In contrast, SOLD w/o OCE employs a deterministic autoregressive transformer-based predictor module that forecasts the next state given all previous past states and the action vector.
> Action/Reward Estimation: DreamerV3 uses MLPs for the actor, critic and reward models, whereas SOLD w/o OCE uses our proposed Slot Aggregation Transformer, albeit using a holistic representation instead of object slots.
> Encoder/Decoder: Both models use convolutional encoder-decoder architectures that map the input images to embedding representations.
>
> **Are tricks like EMA, spatial bins, and imagination also utilised in DreamerV3?**
>
> Yes. EMA, binning and latent imagination are features proposed or utilized in DreamerV3.
> We also implement these tricks into our SOLD framework in order to make a fair comparison with DreamerV3 and validate in isolation the effect of a structured object-centric state representation.
>
> **What is the effect of removing the pre-training stage of the world model?**
>
> Our SOLD framework, similar to other models extending the SAVi decomposition model (Slotformer, OCVP, …), relies on having good object representations in order to learn an accurate world model.
> End-to-end training of of SAVi and RL, i.e. without any pretraining, proved to be challenging in our initial experiments; therefore, similar to related works, we decided to first pretrain SAVi using a small number of episodes.
> We want to emphasize that the SAVi object-centric decomposition is the only part of the world model that is pre-trained, whereas the dynamics and reward models are learned in the RL loop. Furthermore, as shown in Fig. 5, we report that SAVi pretraining only requires a small percentage of the total training time and episodes.
>
> **What’s the performance of DreamerV3 if pre-training its world model instead of learning from scratch?**
>
> DreamerV3 encodes the visual observations into an unstructured holistic state representation using a CNN-based encoder-decoder. The Dreamer algorithms have shown to learn good behaviors without the need for explicit pretraining of the encoder-decoder before RL training.
> However, even when accounting for the SAVi pretraining episodes, SOLD outperforms DreamerV3’s sample efficiency on robotic environments that require object-relational reasoning.

---

> ### Author Response · Authors · 2024-11-19
> **Response to Reviewer W43M (2/2)**
>
> **In Fig. 6, the authors claim that SOLD discovers objects relevant to tasks. However, the object is not discovered until the manipulator begins to interact with it (after t=3).**
>
> We want to clarify that Fig. 6 depicts the normalized attention maps from the actor over object tokens, and not the objects discovered by SAVi.
> This figure does not illustrate SAVi’s ability to parse the input image into object components (this ability is displayed in Fig. 3, where each decoded object slot is given a different color), but rather SOLD’s ability to selectively attend to task-relevant objects over long time horizons. Therefore, the time dimension in the figure represents the actor’s attention at time-step t=21 over its context, which comprises all current and previous object tokens.
> In this example, the actor cannot attend to the representation of the red target object at time-step t=2 as this object is fully occluded. Only on time-step t=3 does the object briefly become visible and thus can the model represent and attend to such an object.
> Other relevant objects (e.g. green cube or robot arm) are visible in the current and recent time-steps and therefore there is no reason to strongly attend to them in earlier time-steps.
> Another example of this effect can be seen in Fig. 16, where the model attends to the red target sphere in past time-steps, since it is occluded in the current one, while focussing on the green cube in the current time-step since it is clearly visible.
> These visualizations highlight which inputs influence the actor's decision-making process, demonstrating an understanding of the relative importance of different objects for solving a given task.
>
> **Can DreamerV3 or SOLD w/o OCE already discover target objects?**
>
> All methods (SOLD, DreamerV3, and SOLD w/o OCE) must encode the input images into a state representation that encodes relevant properties of objects in the scene to solve the task. The main difference lies in the state representation used by the different methods.
> SOLD encodes the input images into a set of object slot embeddings, each corresponding to a different object. This structured representation, in combination with the OCVP predictor and Slot Aggregation Transformer, allows the agent to explicitly model object dynamics and interactions, and learn a proficient policy in a sample-efficient manner.
>
> In contrast, both DreamerV3 and SOLD w/o Slots encode the observed images first into sets of feature maps and then into embedding vectors. The target objects must be encoded into such feature maps and embeddings in order to learn a proficient policy. However, both DreamerV3 and SOLD w/o Slots employ a *holistic representation* of the scene consisting of a single embedding that jointly encodes all the relevant attributes present in the input.
>
> **How do action/reward labels influence the representational capacity of SAVi?**
>
> We would like to highlight that SAVi is able to parse an image into its object-centric components without the need for action or reward information, as shown in Figure 3 at time-step t=0, which represents the context frame for the sequences.
> The remaining time steps depict open-loop predictions obtained by the Conditional-OCVP module, which autoregressively forecasts future object states and frames conditioned on the history of object states and robot actions. We require action commands as conditioning signal since forecasting the robot’s movement is not possible from images alone.
> Rewards and values do not influence the performance of our model for object-centric decomposition or for predicting future frames, and are only needed for behavior learning in latent imagination.
>
> **Would it be possible to combine SOLD with TD-MPC?**
>
> Both TD-MPC and TD-MPD2 are reconstruction-free methods where the internal state is learned solely through task-relevant predictions of reward, value, and action. Our current framework follows the SAVi formulation, which relies on a reconstruction loss in the image space in order to learn object-centric representations, thus deviating from the TD-MPC algorithm.
> However, there is a growing body of literature on object-centric learning without reconstruction (e.g. [1], [2]), which could bridge the gap between SOLD’s framework and the TD-MPC model, and provides a good future research avenue.
>
> [1] Baldassarre et al.: Towards Self-Supervised Learning of Global and Object-Centric Representations, ICLR-W 2022.
>
> [2] Löwe et al.: Learning object-centric video models by contrasting sets: NeurIPS-W 2020.
>
> **Does the learned model show robustness to scene disturbances?**
>
> To assess the robustness of the learned policies to changes in the scene, we performed an additional experiment that replaces the original set of 16 colors for blocks and targets with a new set of 32 colors. We have added the results of this study in the App. F.4. We are currently running further generalization experiments for all models on our suite of robotic tasks for the final version.

---

> > ### Comment · Reviewer_W43M · 2024-11-22
> >
> > Thanks for the response and additional results. I do think the idea of object-centric latent dynamics is promising to make MBRL more generalizable. While many questions have been well answered, I still wish to see more experimental results to verify this idea. I would like to suggest the authors (1) compare with more SOTA algorithms like TD-MPC2, as recognized by other reviewers as well, (2) include more complex and visually realistic simulations, like ManiSkill2, MyoSuite or DexArt, and, if possible, (3) real-world manipulation tasks with offline MBRL.

---

### Official Review · Reviewer_TMTo · 2024-11-03

**Soundness:** 2
**Presentation:** 3
**Contribution:** 2
**Rating:** 5
**Confidence:** 3

**Summary:**

This paper introduces SOLD, a model-based reinforcement learning approach that learns object-centric dynamics from pixel inputs without supervision. SOLD models the world as distinct objects with interactions, inspired by human reasoning. This object-focused latent space improves interpretability and supports more efficient skill learning. Results show that SOLD outperforms DreamerV3 in robotic tasks that require both understanding relationships and precise manipulation, demonstrating the benefits of object-centric learning in reinforcement learning.

**Strengths:**

1. This work provides an object-centric solution for world-model training based on Slot Attention, and applies it into the following behavior learning.
2. The paper is well-written and easy to follow.

**Weaknesses:**

1. Missing baselines from the experiment part. More model-based RL methods beyond DreamerV3 such as TD-MPC [1], TD-MPC2 [2] can be competitive and representative baselines for this work. Additionally, more related works on model-based RL can be added.
2. Limited task domain. The paper mainly focuses on reach and pick-place tasks, which has limited diversity in the task domain. Object-centric tasks with more skills can be added, such as open/close/pull/push articulations, which poses more challenge on as object-centric representation and skill learning.

[1] Hansen N A, Su H, Wang X. Temporal Difference Learning for Model Predictive Control[C]//International Conference on Machine Learning. PMLR, 2022: 8387-8406.
[2] Hansen N, Su H, Wang X. TD-MPC2: Scalable, Robust World Models for Continuous Control[C]//The Twelfth International Conference on Learning Representations.

**Questions:**

1. I'm curious about the application of the representation on multi-task, since object-centric and task-agnostic representations can be smoothly incorporated into a multi-task setting.
2. The authors have shown fine-tuned SAVi can capture unseen state distribution by visulization of masked reconstruction. Please further demonstrate whether the fine-tuned SAVi can benefit RL training.
3. The authors only demonstrate final scores of the non-object-centric baselines. More curves, comparison with baselines and visualizations can be added for clarification.

---

> ### Author Response · Authors · 2024-11-19
> **Response to Reviewer TMTo**
>
> We thank the reviewer for the constructive review and for acknowledging the clarity of our paper! Following the feedback, we added new sections to the appendix including the following:
> - review of further Model-Based RL related works (App. B),
> - training curves depicting the success-rate and return of our proposed SOLD model on the DM-Control and MetaWorld environments (App. F.3), and
> - an ablation study evaluating the importance of SAVi fine-tuning during RL training (App. F.5).
> We are currently extending this ablation with quantitative results on the RL performance by training the RL model w/o fine-tuning SAVi for 3 seeds on each of the PickAndPlace tasks.
> Below we address further comments:
>
> **I'm curious about the application of the representation on multi-task, since object-centric and task-agnostic representations can be smoothly incorporated into a multi-task setting.**
>
> That is a very interesting question.
> Object-centric learning methods often require reasonably large datasets in order to extract object slots in an unsupervised manner. A multi-task setting is inherently aligned with this paradigm, providing more data and of increased diversity, which could be exploited to learn a better state representation.
> We do not apply our model on a multi-task RL setting in this work. Investigating if object-centric representations can lead to improved sample efficiency in multi-task scenarios is a promising idea for future work.
>
> **The paper mainly focuses on reach and pick-place tasks, which has limited diversity in the task domain. Object-centric tasks with more skills can be added.**
>
> We understand the reviewer’s concerns about the task diversity. However, we slightly disagree with this premise as, even though we do not address articulate tasks (e.g. open/close), we do evaluate our SOLD framework on environments such as MetaWorld-Hammer and DM-Control-Finger-Spin, which are more complex than simple reach/pick-and-place tasks.
> We agree that integrating more challenging manipulation tasks with relational reasoning, such as picking realistic items from clutter or learning behaviors for more complex manipulators are exciting future research directions.

---

> > ### Comment · Reviewer_TMTo · 2024-11-21
> >
> > Thanks for your reply! While most of my concerns are clarified, I still think more model-based RL baseline results such as TD-MPC2 are necessary, which is also mentioned by other reviewers. I am willing to raise my rating if more results can be posted.

---

### Official Review · Reviewer_MUQR · 2024-11-08

**Soundness:** 3
**Presentation:** 3
**Contribution:** 2
**Rating:** 5
**Confidence:** 4

**Summary:**

This paper presents SOLD, a model-based RL method designed to learn object-centric representations directly from pixels. SOLD’s focus on structuring the latent space around individual objects and their interactions is a novel take that could be especially useful for tasks like robotic manipulation, where reasoning about individual objects is crucial. Experimental results show that SOLD performs strongly, particularly in object manipulation tasks that require relational reasoning between objects from Meta-World.

**Strengths:**

- By breaking down the environment into individual objects, the model can reason about complex interactions more intuitively, which has clear benefits for manipulation and relational tasks.

- The paper is well-organized, and the authors do a nice job of explaining SOLD’s inner workings. Visualizations of attention patterns and slot decomposition are particularly helpful, showing how SOLD parses and tracks key objects in the scene.

- The experiments show promising adaption results state distributions and performance on multi-object tasks.

**Weaknesses:**

- While SOLD performs well on DM-Control and Meta-World, its robustness would be clearer if evaluated across a broader set of RL benchmarks, such as Atari, ProcGen, or Minecraft, as well as in complex robotics environments like ManiSkill2, IsaacGym, and Robosuite. Expanding the scope of environments could better showcase SOLD’s versatility.

- An ablation study on the slot attention mechanism by replacing it with another transformer-based model rather than a CNN-based encoder-decoder would help clarify its unique impact on SOLD’s object-centric understanding and performance.

- The Slot Aggregation Transformer likely plays a significant role in capturing relational dynamics. Testing alternative transformer configurations could reveal how essential this component is for tasks that require multi-object relational reasoning.

**Questions:**

- Have you considered ablations to evaluate the importance of specific components, such as removing slot attention, testing alternative aggregation methods for slots? This would clarify the role of each part in SOLD’s overall performance.

- Is there a plan to test SOLD on more benchmarks, like Control Suite, Atari, or more demanding robotics tasks like ManiSkill2 or IsaacGym? These benchmarks would give a clearer sense of SOLD’s versatility and adaptability.

- Could you share detailed return scores on DM-Control?

- How did you determine the slot dimension? In the visualization of the decomposition results, a small number of slots might be enough for the various tasks you presented.

---

> ### Author Response · Authors · 2024-11-19
> **Response to Reviewer MUQR**
>
> We thank the reviewer for the detailed comments and constructive feedback! We very much appreciate positive comments about our paper and experiments. Below we address your questions:
>
> **Have you considered ablations to evaluate the importance of specific components, such as removing slot attention, testing alternative aggregation methods for slots? This would clarify the role of each part in SOLD’s overall performance.**
>
> We have indeed considered some ablation studies in our experiments.
> We compared our SOLD model with a non-object-centric variant, thus evaluating the importance of a structured object-centric state with respect to a single holistic representation. This non-object-centric baseline can be seen as removing Slot Attention from our method.
> Furthermore, we qualitatively ablate the effect of fine-tuning SAVi for learning the behavior, showcasing the inability of the model with a frozen SAVi encoder to capture state distributions unseen during pretraining, e.g. lifting a block off the table or grasping the hammer. We have added a second visualization in appendix F.5. We are currently extending this ablation with quantitative results on the RL performance by training the RL model w/o fine-tuning SAVi for 3 seeds on each of the PickAndPlace tasks.
>
> **Could you share detailed return scores on DM-Control?**
>
> We have added section F.3 to the appendix including training curves depicting the success-rate and return on the DM-Control and Meta-World environments.
>
> **Is there a plan to test SOLD on more benchmarks, like Control Suite, Atari, or more demanding robotics tasks like ManiSkill2 or IsaacGym?**
>
> Our focus in this work is to demonstrate that it is possible to perform object-centric MBRL directly from pixels and that it can improve relational reasoning abilities of the resulting agent.
> Furthermore, we also show that our object-centric SOLD method can also learn performant agents on non-object-centric environments, such as those from DM-Control and Meta-World.
> To apply SOLD on more challenging robotics tasks, we are limited by existing object-centric methods, which often fail to generalize to visually complex environments. However, our proposed framework is general and could be updated to integrate more recent and capable object-centric decomposition models, which could allow SOLD to be applicable to more challenging robotic environments. We indeed plan to address this in future work.
>
> **How did you determine the slot dimension? In the visualization of the decomposition results, a small number of slots might be enough for the various tasks you presented.**
>
> We set the dimensionality of the slot embeddings to 128, which we empirically find a suitable value for learning accurate object-centric representations.
> Regarding the number of object slots used, following common practice in the object-centric learning literature, we set this hyper-parameter as the maximum number of objects that can be present simultaneously in an observation. For instance, in a PickAndPlace task we employ 10 slots: one for the background, one for the robot, two for the gripper fingers, one for the target location, and five for the blocks.
> A summary of the number of slots used in the datasets, as well as the object they represent, is as follows:
>
> | **Environment**       | **Number of Slots** | **Objects in the Scene**                             |
> |------------------------|:-------------------:|-----------------------------------------------------|
> | DM-Cartpole           |          2          | Cartpole and background                             |
> | DM-FingerSpin         |          3          | Finger, spinner and background                     |
> | MW-ButtonPress        |          4          | Button box, background, and two slots for the robot arm |
> | MW-Hammer             |          6          | Nail box, background, hammer, and three slots for the robot arm |
> | PickAndPlace          |         10          | Five blocks, target, background, robot arm, and two gripper fingers |
> | Push                  |          8          | Five blocks, target, background, and robot arm      |
> | Reach                 |          8          | Five blocks, target, background, and robot arm      |

---

> ### Comment · Reviewer_MUQR · 2024-11-22
>
> Thank you for your response and clarification. I believe learning object-centric dynamics from pixel inputs is valuable for MBRL and can potentially make it more generalizable and interpretable. However, I agree with the other reviewers that additional experiments across diverse environments—such as ManiSkill2, IsaacGym, and Robosuite, as mentioned in my comments—along with incorporating different viewpoints, would provide deeper insights into the methods and their applicability.

---

### Official Review · Reviewer_JsXm · 2024-11-08

**Soundness:** 2
**Presentation:** 3
**Contribution:** 2
**Rating:** 3
**Confidence:** 4

**Summary:**

This paper presents a new model-based RL algorithm from image observations called Slot-Attention for Object-centric Latent Dynamics (SOLD). The main novelty compared to existing MBRL methods like Dreamer is a latent space structure which promotes the learning of object-centric representations. Specifically, SOLD builds on the Slot Attention mechanism used in SAVi and OCVP (video prediction only, no control) to build the world model. The rest of the MBRL pipeline is built around the Slot Attention representations similar to the settings in DreamerV3.

The authors evaluate the SOLD algorithm mainly on a custom-made multi-object environment built on the Gym Fetch simulator. The benchmark consists of reach, push and pick-and-place tasks where multiple objects are present in the scene. Aside from the basic setting to manipulate a specific color, other more challenging variants include choosing a distinct color within the collection and choosing the reddest one among similar colors.

Visualization of open-loop predictions demonstrate that the Slot Attention architecture effectively identifies independent moving parts in the images, including both manipulatable objects and robot body links. SOLD also outperforms DreamerV3 in policy learning in terms of sample efficiency and final policy performance. Aside from the custom benchmark specifically designed for multi-object reasoning, the authors also demonstrate the applicability of SOLD in more main-stream benchmarks including DM Control and Meta-World.

**Strengths:**

- The idea of learning a latent world model with an object-centric representation is sound and extending the known Slot Attention mechanism to a control setting is a good move.
- The authors created a custom-made task suite to highlight the utility of object-centric reasonings in a MBRL setting. In these tasks, SOLD outperforms the state-of-the-art MBRL method DreamerV3.
- The main method is explained pretty clearly by the equations and system diagrams.
- The visualization of the slots and attentions show that the model indeed captures the moving objects in the Fetch environment as well as Meta-World and DM Control.

**Weaknesses:**

- The method is showcased mainly on a custom-made environment, where the settings are a bit detachched from reality. I think this work can shine in a cluttered manipulation environment with realistic objects. The authors touches on Meta-World and DM Control tasks, but no systematic RL comparison is included.
- The core contribution of this work seems to be only adding actions to OCVP and putting it in an RL loop. If this is the case, the contribution is a bit limited, unless more systematic studies or hardware experiments are conducted.

**Questions:**

- What is the inference latency of SOLD's latent world model? Dreamer's RSSM is fairly light weight.
- Aside from the parameter count, as outlined in the appendix, are there any other main difference between SOLD w/o OCE and DreamerV3? I wonder where the performance gain comes from.
- Is SOLD more sample efficient in Meta-World and DM Control tasks too?
- Could there be some special treatments if more than one camera view is used?

---

> ### Author Response · Authors · 2024-11-19
> **Response to Reviewer JsXm**
>
> Thank you for your review and constructive comments! We appreciate that you recognize the strengths of our method, including the idea of an object-centric world model for model-based RL, the performance improvement of our approach with respect to DreamerV3, as well as the clarity in presentation. Below we address the highlighted weaknesses and questions:
>
> **This work can shine on cluttered manipulation of realistic objects.**
>
> We agree that good object-representations could help learn good policies in a realistic cluttered manipulation environment. However, learning accurate object-centric representations of visually challenging scenes from videos still remains a challenge.
> Nevertheless, our proposed framework is general and could be updated to integrate more recent and capable decomposition models, allowing SOLD to be applicable to real world videos. We plan to address this challenge in future work.
>
> **No RL comparison on Meta-World and DM Control.**
>
> We have added section App. F.3 including return and success-rate curves of those environments.
>
> **Core contribution seems to be only adding actions to OCVP and putting it in an RL loop.**
>
> We agree that extending OCVP to an action-conditional setting and using it as a world-model is a core part of our paper. To the best of our knowledge, we are the first to propose a fully object-centric model-based RL (MBRL) framework that learns directly from pixels.
> Previous works like SMORL [1] or ETH [2] combined object-centric representations with model-free RL, whereas methods like FOCUS [3], despite exploiting object slots at some stage of the MBRL pipeline, still rely on a holistic latent representation of the scene. In contrast, we propose a complete MBRL pipeline using unsupervised object slots.
>
> Using object-centric representations in the RL framework presented several non-trivial challenges. We believe that our solutions are of interest to the community. For instance, whereas most works freeze the object-centric decomposition model (e.g. SAVi) after pretraining, our model is among the first proposed methods to fine-tune it so as to adapt to continually changing state distributions, which is required to learn performant policies.
> Moreover, we design our Slot Aggregation Transformer to make predictions by aggregating information from a history of object slots.
>
> **What is the inference latency of SOLD's latent world model?**
>
> For a batch size of 64 on a RTX 4090 GPU, the forward pass of the latent dynamics model takes 5.9869 ms, while it takes 9.6243 ms to encode the images into slots with SAVi. Therefore, our model is more than real-time capable.
>
> **Differences between SOLD w/o OCE and DreamerV3.**
>
> Our SOLD w/o OCE corresponds to a variant of SOLD, but replacing our structured state composed of multiple object slots, with a single holistic embedding representation.
> Therefore, some architectural components in SOLD w/o OCE and DreamerV3 present fundamental differences:
> - **Predictor:** DreamerV3 uses a stochastic recurrent RSSM world model implemented with Block-GRU modules. In contrast, SOLD w/o OCE employs a deterministic autoregressive transformer-based predictor module that forecasts the next state given all previous states and the action vector.
> - **Action/Reward Estimation:** DreamerV3 uses MLPs for the actor, critic and reward models, whereas SOLD w/o OCE uses our proposed Slot Aggregation Transformer, albeit using a holistic representation instead of object slots.
> - **Encoder/Decoder:** Both models use convolutional encoder-decoder architectures that map the input images to embedding representations.
>
> **Is SOLD more sample efficient on Meta-World and DM Control?**
>
> The structured object-centric state employed by SOLD does not yield large benefits for environments such as those from Meta-World or DM-Control, where reasoning about objects is not needed to solve the tasks. Therefore, SOLD cannot match the optimized efficiency of DreamerV3.
> Nevertheless, we highlight that SOLD is a general framework that outperforms SOTA on object-centric tasks, which are common in robotics contexts, while still being performant on environments beyond its intended application.
>
> **Extension to multiple camera views.**
>
> This is an interesting future research direction. If multiple views were available, we could replace SAVi with a Multi-View object-centric decomposition model (e.g OSRT [4]), thus learning 3D-aware object slots, which could lead to more accurate state predictions.
> Furthermore, using multiple input views would reduce the amount of occlusions thus simplifying the task.
>
>
> [1] Zadaianchuk et al.: Self-supervised visual reinforcement learning with object-centric representations, ICLR 2021.
>
> [2] Haramati et al.: Entity-centric reinforcement learning for object manipulation from pixels,  ICLR 2024.
>
> [3] Ferraro et al.:  FOCUS: Object-centric world models for robotic manipulation, NeurIPS-WS 2024.
>
> [4] Sajjadi et al: Object Scene Representation Transformer, NeurIPS 2022.

---

> > ### Comment · Reviewer_JsXm · 2024-11-22
> >
> > Thanks for answering my questions and sharing additional details. I genuinely believe that getting object-centric representations to work with MBRL is non-trivial and valuable work. However, I agree with other reviewers that more experiments are required to justify the approach. For one, SOLD doesn't compare favorably to DreamerV3 in environments other than the custom ones. Also, comparison with methods like TransDreamer and TD-MPC can help better evaluate this work.

---

### Author Response · Authors · 2024-11-19
**General Comment to Reviewers**

We thank the reviewers for their comments and constructive feedback, which has helped us improve the paper!

We are happy that reviewers acknowledge that a structured object-centric model-based RL framework is a good solution (Reviewers JsXm, MUQR, TMTo), that our paper is clear and well organized (Reviewers JsXm, MUQR, TMTo, W43M), that our visualizations depict interpretable internal representations (Reviewers JsXm, MUQR), and the strong performance of our method on our robotics task suite (Reviewers JsXm, W43M).

Based on reviewer feedback, we have updated our submission by adding new sections to the appendix (highlighted in blue color) including the following content:
- A more thorough review of Model-Based RL related works (App. B),
- Training curves depicting the success-rate and return of our proposed SOLD model on the Meta-World and DM-Control environments, respectively (App. F.3).
- An experiment evaluating the generalization capabilities of different RL models to objects of colors unseen during training (App. F.4).
- A qualitative ablation study evaluating the importance of SAVi fine-tuning during RL training (App. F.5).

We intend to provide further ablation studies, which are currently being trained, regarding model generalization, as well as a quantitative ablation study of RL performance of models without SAVi fine-tuning.

We address further comments below in direct replies to the reviews.

---

### Meta-Review · Area_Chair_pz5m · 2024-12-25

**Metareview:**

The paper introduces SOLD (Slot-Attention for Object-centric Latent Dynamics), an unsupervised algorithm that learns object-centric dynamics from pixel inputs, enhancing interpretability and efficiency in model-based reinforcement learning, and demonstrating superior performance to DreamerV3 in robotic benchmarks focused on relational reasoning and manipulation.

While the reviewers acknowledged the paper's contributions, they raised significant concerns: (1) the absence of comparisons with additional essential baselines, (2) evaluations limited to a narrow set of environments, and (3) questions about the proposed method's performance relative to DreamerV3 in environments beyond the custom ones used in the paper. The AC recommends that the authors carefully address these concerns to strengthen the paper in a future revision.

**Additional Comments On Reviewer Discussion:**

As the reviewers were unanimous in their recommendation, no substantial discussion occurred during the Reviewer Discussion phase.

---

### Decision · Program_Chairs · 2025-01-22

Reject